# Towards accurate methane point-source quantification from high-resolution 2D plume imagery

Siraput Jongaramrungruang[1], Christian Frankenberg[1,2], Georgios Matheou[3], Andrew K. Thorpe[2], David R. Thompson[2], Le Kuai[4], Riley M. Duren[2]

[1]Division of Geological and Planetary Sciences, California Institute of Technology, Pasadena, CA 91125, USA
[2]NASA Jet Propulsion Laboratory, California Institute of Technology, Pasadena, CA 91109, USA
[3]Department of Mechanical Engineering, University of Connecticut, Storrs, CT 06269, USA
[4]Joint Institute for Regional Earth System Science and, University of California, Los Angeles, CA 90095, USA

*Correspondence to*: Siraput Jongaramrungruang (siraput@caltech.edu); Christian Frankenberg (cfranken@caltech.edu)

**Abstract.** Methane is the second most important anthropogenic greenhouse gas in the Earth climate system but emission quantification of localized point sources has been proven challenging, resulting in ambiguous regional budgets and source categories distributions. Although recent advancements in airborne remote sensing instruments enable retrievals of methane enhancements at unprecedented resolution of 1-5 m at regional scales, emission quantification of individual sources can be limited by the lack of knowledge of local wind speed. Here, we developed an algorithm that can estimate flux rates solely from mapped methane plumes, avoiding the need for ancillary information on wind speed. The algorithm was trained on synthetic measurements using Large Eddy Simulations under a range of background wind speeds of 1-10 m s$^{-1}$ and source emission rates ranging from 10 to 1000 kg h$^{-1}$. The surrogate measurements mimic plume mapping performed by the next generation Airborne Visible/Infrared Imaging Spectrometer (AVIRIS-NG) and provide an ensemble of 2-D snapshots of column methane enhancements at 5 m spatial resolution. We make use of the integrated total methane enhancement in each plume, denoted as Integrated Methane Enhancement (IME), and investigate how this IME relates to the actual methane flux rate. Our analysis shows that the IME corresponds to the flux rate non-linearly and is strongly dependent on the background wind speed over the plume. We demonstrate that the plume width, defined based on the plume angular distribution around its main axis, provides information on the associated background wind speed. This allows us to invert source flux rate based solely on the IME and the plume shape itself. On average, the error estimate based on randomly generated plumes is approximately 30% for an individual estimate and less than 10% for an aggregation of 30 plumes. A validation against a natural gas controlled-release experiment agree to within 32%, supporting the basis for the applicability of this technique to quantifying point sources over large geographical areas in airborne field campaigns and future space-based observations.

# 1 Introduction

Methane is the second most important anthropogenic greenhouse gas in Earth's atmosphere, with additional indirect impacts as it affects both tropospheric ozone and stratospheric water vapor. Despite its significance, our understanding of global and regional $CH_4$ budgets has remained inadequate due to the fact that the strength and distribution of $CH_4$ emissions from various source types are not well-constrained (Houweling et al., 2017; Turner et al., 2017). Estimates of $CH_4$ emissions from point sources (e.g. at facility scale) are particularly uncertain, since space-based observations lack sufficiently fine spatial resolutions while in situ measurements are too sparse and mostly representative of large-scale background concentrations. Improved estimates of the $CH_4$ emissions at this point-source scale are critical in guiding emission mitigation efforts.

Recent developments in airborne imaging spectroscopy techniques to quantify $CH_4$ plumes have opened the way for $CH_4$ measurements at sufficiently high spatial resolution needed to differentiate various local sources within regional scales (Frankenberg et al., 2016; Hulley et al., 2016; Thompson et al., 2015; Thorpe et al., 2016a, 2017; Tratt et al., 2014). A recent airborne campaign in the Four Corners region retrieved column methane enhancements at a resolution of 3 m (Frankenberg et al., 2016), enabling the observation of the plume shape in direct vicinity of the point source. During the campaign, many plumes of various sizes ranging from a few tens of meters to hundreds of meters were detected across the region, with the majority of their source emission rates between 10 and 1000 kg($CH_4$) h$^{-1}$ (Frankenberg et al., 2016). This allows for an effective way to remotely identify and locate $CH_4$ emissions from point sources such as pipeline leaks or oil and gas facilities. The retrievals provide the quantification of a column enhancement (e.g. in molecule cm$^{-2}$ above background), which can be integrated across the entire methane plume to derive the total amount of methane within the plume, denoted as Integrated Methane Enhancement (IME, either in molecule or mass units, Frankenberg et al., 2016). In addition, the instrument observes the fine structure of the plume at an unprecedented spatial resolution. However, the flux inversion from the observed plumes to the actual emission rate at the source remains complicated due to the dependence on tropospheric boundary layer conditions such as wind speed and atmospheric stability during the overpass. To interpret the relationship between the observed plumes and flux rates, previous studies have relied on Gaussian plume inversion models (Krings et al., 2011, 2013; Rayner et al., 2014; Nassar et al., 2017; Schwandner et al., 2017) or an airborne in-situ approach using a mass balance calculation based on the enhancement downwind of the source (Cambaliza et al., 2015; Conley et al., 2016; Gordon et al., 2015; Jacob et al., 2016; Lavoie et al., 2015). Frankenberg et al. (2016) used a simple linear scaling between IME and flux rate, which allowed for a straightforward derivation of fluxes from the observed IME given an averaged wind speed across a large region for the campaign over several days. Varon et al., 2018 estimated flux rate as IME divided by the residence time of methane in the plume calculated based on the effective length of the plume from its area and the effective wind speed inferred from 10 m wind speed by in situ measurement or meteorological reanalysis data. All of these methods rely on knowledge of local wind speed, which is acquired through either in situ wind measurements or the estimation from meteorological forecast or reanalysis data. The former can be costly and time consuming without prior knowledge of source locations, while the latter can be

inaccurate due to the rapid changes of a local plume over a much shorter temporal and spatial scale (minutes, hundreds of meters) than the typical atmospheric reanalysis products (a-few-hourly average, tens of kilometers).

In this work, we aim to improve our understanding of how the inferred emission rates change under different atmospheric conditions, e.g. the errors due to a lack of accurate wind measurements. To investigate this relationship and associated errors, we used Large Eddy Simulations (LES, Matheou and Bowman, 2016) to simulate the plume dynamics at high spatial resolution (5 m) with prescribed source rates under various background wind speeds and typical surface latent and sensible heat fluxes. Using 3D LES model output for each snapshot, we simulated synthetic 2D airborne measurements by applying the respective averaging kernels. Based on these synthetic measurements, we developed an algorithm to deduce the wind speed from the plume's spatial distribution and investigate the degree to which the flux rate can be inverted from only the remotely-sensed $CH_4$ retrievals. This allowed us to perform an end-to-end test of errors in inverted methane fluxes in both the absence and presence of ancillary information on the actual wind speed (Section 6.3).

This work was inspired by the use of IME to quantify methane single-point sources from field campaigns using airborne instruments. These plumes generally are of small-to-medium sizes (<2 kilometers). The concept, nevertheless, can be applicable to larger sources as well as toward measurement of localized sources from space in the coming decade for satellite retrievals at a much finer spatial resolution (Thorpe et al., 2016b).

Section 2 illustrates the plume observations and the instrument specifications. Section 3 will give a brief overview of Gaussian plume modelling. The setup of the LES and application of instrument operators to simulate airborne measurements are described in Section 4 and Section 5 respectively. Section 6 shows simulated plumes under different atmospheric scenarios and the relationship between observed IME and actual emission rates. The error analysis of flux inversion based on the IME method is also provided. The final section provides a discussion and conclusion.

## 2 Plume Observations and Instrument Specifications

Fig. 1 shows examples of observed methane plumes using the next-generation Airborne Visible/Infrared Imaging Spectrometer (AVIRIS-NG) and the Hyperspectral Thermal Emission Spectrometer (HyTES) during the Four Corners flight campaign (Frankenberg et al., 2016). The Iterative Maximum a Posteriori Differential Optical Absorption Spectroscopy (IMAP-DOAS) method (Thompson et al., 2015) and Clutter Matched Filter (CMF) were used to retrieve the scenes from AVIRIS-NG and HyTES respectively. In this case, the aircraft repeatedly flew over a coal mine venting shaft, with approximately 10 minute revisit time. Evidently, the plume is changing in time and exhibits fine-scaled features due to atmospheric turbulence. Quantifying the source rate from detected plumes using atmospheric simulations to understand their behavior and variations in space and time is the main subject of this work. In order to compare our simulations with actual observations, we need to take the measurement characteristics of the remote sensing instrument into account. This relates to both measurement precision, which determines detection thresholds which marks and defines the detected plume, as well as vertical sensitivity, which affects what parts of the plume structure can actually be observed. Depending on the techniques being used, both can vary widely.

The left column in Fig. 1 shows scenes that are retrieved from the AVIRIS-NG instrument, which measures reflected solar radiation between 0.35 $\mu$m and 2.5 $\mu$m at 5 nm resolution and sampling (Hamlin et al., 2011; Thompson et al., 2015). To first order, it has a uniform vertical sensitivity (averaging kernel) of 1 at each height (see Fig. 2). Another instrument that was used in the Four Corners campaign is HyTES, which enables the detection of $CH_4$ plumes due to its absorptions in the thermal infrared around 7.65 $\mu$m (Hulley et al., 2016). Its varying sensitivity in the vertical can be calculated as the derivative of the retrieved total column amount with respect to the change in a particular layer. These vertical sensitivities are formally called column averaging kernels. They inform us how well methane deviations from the prior at each height can be measured, which determines whether they will be visible in retrieved column enhancements. Mathematically, we can express this relationship as

$$E(i,j) = \sum_k (\Delta x \Delta y \Delta h) * C(i,j,k) * CAK(k) \tag{1}$$

where $E(i,j)$ is the observed total column enhancement (mass or molecules) at the horizontal grid cell $(i,j)$. $\Delta x, \Delta y$ and $\Delta h$ are grid sizes in $\hat{i}, \hat{j}$ and $\hat{k}$ respectively, $C$ is the concentration (mass or molecules per volume), $CAK(k)$ denotes the column averaging kernel evaluated at level $k$. Technically, the $CAK$ can also be a function of location $(i,j)$ but for the purpose of producing synthetic measurements from our simulations in this work, we apply the $CAK$ only as a function of height.

Fig. 2 illustrates the difference between the column averaging kernels that we use to model AVIRIS-NG and HyTES synthetic measurements. The distinct column averaging kernels of both instruments hold significant importance, each with its advantages and disadvantages. The column averaging kernel of AVIRIS-NG is approximately uniform across all vertical levels, which implies that the retrieved column enhancement accurately reflects the actual column enhancement. On the other hand, the sensitivity of HyTES is almost zero near the surface but increases with height, becoming even larger than 1 at a certain height. This means that the instrument is almost blind to methane near the ground, but amplified the actual methane amount at certain heights in the column. This distinction is evident in Fig.1 where the observed methane plume remains more consistent from AVIRIS-NG scenes, whereas more variations appear in the HyTES scenes potentially due to changes in plume vertical structures. It should also be noted that the HyTES averaging kernel strongly depends on the temperature profile as well as the surface temperature, which can vary within and between scenes. In contrast, averaging kernels using short-wave reflected light are less variable.

## 3 Gaussian Plume Modelling and Its Limitations

The simplest way to simulate plumes is Gaussian plume modelling, which assumes a steady and uniform wind $U$ along the x-axis and orthogonal spreading of the plume in crosswind (y-axis) and vertical (z-axis) directions. The spreading of the plume depends on the dispersion functions $\sigma_y(x)$ and $\sigma_z(x)$. The dispersion functions depend on the atmospheric stability (Pasquill, 1961). For instance, convective conditions favor vertical dispersion, whereas in a stable atmosphere the plume primarily disperses in the horizontal directions (Briggs, 1973; Matheou and Bowman, 2016; Sutton, 1931). The three-dimensional Gaussian plume equation is given by (Matheou and Bowman, 2016)

$$C(x, y, z) = \frac{1}{2\pi\sigma_y(x)\sigma_z(x)} \cdot \frac{Q}{U} \cdot \exp\left[\frac{-y^2}{2\sigma_y^2(x)}\right] \sum_{m=0}^{\infty} \left( \exp\left[-\frac{(z - 2mz_i)^2}{2\sigma_z^2(x)}\right] + \exp\left[-\frac{(z + 2mz_i)^2}{2\sigma_z^2(x)}\right] \right) \qquad (2)$$

where $C(x, y, z)$ is the (equilibrium) concentration at each point in the 3-dimensional space within the atmospheric boundary layer with inversion height $z_i$. The model assumes a reflective boundary condition where the parameter m multiplied by $z_i$ indicates the height that the reflection occurs and the summation over this parameter m represents the equivalent

5   concentration within 0 to $z_i$. $Q$ is the source flux rate at the origin. The variances $\sigma_y(x)$ and $\sigma_z(x)$ are given by empirical relations based on atmospheric stability following the Pasquill classification (Matheou and Bowman, 2016; Pasquill, 1961).

By integrating Eq. 2 in z-direction, the methane column enhancement can be modelled in analytical form as

$$\bar{C}(x, y) = \frac{1}{\sqrt{2\pi}\,\sigma_y(x)} \cdot \frac{Q}{U} \cdot \exp\left[\frac{-y^2}{2\sigma_y^2(x)}\right] \qquad (3)$$

Based on this model, we can vary source rate, wind speed, and stability category to simulate the 2D integrated concentration

10  field. We then apply a device detection threshold to illustrate how the synthetic Gaussian plume column enhancement may change under distinct atmospheric conditions. Examples of the simulated Gaussian plumes with a flux rate of 300 kg h$^{-1}$ are shown in Fig. 3. The left column of Fig. 3 shows the Gaussian plumes under different wind speeds for a fixed stability category, while the right column demonstrates those under a fixed wind speed at 4 m s$^{-1}$ but different stability regimes.

The wind speed $U$ influences the column enhancement, which, based on Eq. 1, is proportional to the ratio $Q/U$. Thus,

15  the Gaussian plume model suggests a strong dependence of the IME on wind speed, which in turn does not explicitly affect the shape of the plume. One way of quantifying a plume shape is using an aspect ratio in the x-y plane. In the Gaussian plume model, the aspect ratio of the plume only changes when the stability switches from one category to another. Thus, the wind speed is only implicitly linked to the shape of the plumes by affecting the stability categories and changing the crosswind variances (as can be seen in Eq. 3).

20  The stability categories in this model, nonetheless, are based on empirical formulae. In reality, the wind speed can influence the shape and distribution of the plumes more directly through advection of the tracer along the flow. The actual plume observations from the Four Corners campaign (Fig. 1) demonstrate that the plumes are of turbulent nature - at times being discontinuous - and cannot be modelled as Gaussian when only one plume snapshot in time is recorded. Therefore, we utilize an LES model, which yields realistic realization of the turbulent flow and the methane plume, to quantify the effect of

25  wind speed on the plume structure.

**4 Large Eddy Simulation setup**

Realistic modeling of $CH_4$ plumes is a prerequisite for this study. We use LES to model the time-resolved 3-dimensional $CH_4$ distribution in the boundary layer under different atmospheric conditions at resolutions currently available

from aircraft measurements (1-5 m). The LES model setup for the simulation of plumes emanating from point sources is as described in Matheou and Bowman (2016). Further details of the model formulation, including the turbulence parameterization, are in Matheou and Chung (2014). A methane surface point source with a specific emission rate in a cloud-free convective atmospheric boundary layer is simulated. The buoyancy of methane is currently being ignored – a good approximation for the present methane concentrations away from the source.

The atmospheric boundary layer is initialized with a mixed layer inversion free troposphere with an initial inversion height $z_i = 800$ m. The initial potential temperature and specific humidity in the mixed layer are $\theta = 298$ K and $q_t = 6.6$ g kg$^{-1}$. The lapse rate is $\Delta\theta/\Delta z = 0.12$ K m$^{-1}$. The flow in the boundary layer is driven by a constant geostrophic wind in the $x$-direction, $u_g$. Different values of the geostrophic wind from 1 to 10 m s$^{-1}$ are used. The surface sensible and latent heat fluxes are 400 and 40 W m$^{-2}$. These values are based on typical field campaign data. Additional simulations with other sensible and latent heat fluxes are also performed later in Section 6.4. Surface momentum fluxes are estimated using Monin–Obukhov similarity theory (MOST).

The model domain is 10.24 x 2.56 x 1.5 km$^3$ in the $x$, $y$, and $z$ direction and the grid resolution is uniform and isotropic $\Delta x = \Delta y = \Delta h = 5$ m. The model computational time-step is one second. Following one hour of model "spin up", where fully-developed three-dimensional turbulence is established in the boundary layer, the three-dimensional concentration at each location at one-minute intervals (snapshots are written out at every minute) is used to construct the synthetic observations. Furthermore, the 10 m and 2 m wind speeds are extracted from the model output to compare with the large-scale geostrophic wind value in each run.

**5 Synthetic Measurement**

With the output from the LES simulations, we can create synthetic measurement of a plume instance that would enable simulation of observations from any instrument. The procedure is that we apply vertical integration as described by Eq. 1 to the 3-D concentration at a given time step, using the column averaging kernel of the instrument of interest. We apply the column averaging kernel of AVIRIS-NG as well as that of HyTES to produce synthetic measurements for these instruments. The detection thresholds of AVIRIS-NG and HyTES instruments can potentially be dependent on the surface properties such as surface reflectance and surface temperature respectively. However, given the typical scale of the plumes of our interest, we assume an average uniform detection threshold across the scene. Here, we use a constant threshold of 500 ppm-m (or about $1.34 \ 10^{18}$ molecules cm$^{-2}$), which is a common value for AVIRIS-NG. As for HyTES, we used the same threshold to exemplify the differences due to averaging kernels only, as opposed to thresholds. This allows us to understand to what extent each instrument can detect CH$_4$ plumes under various wind speeds.

## 6 Results

The output from the LES run provides a more realistic simulation, compared to the Gaussian model, of the plume dynamics as shown in Fig. 4 for AVIRIS-NG synthetic measurements. The left column of Fig. 4 shows single snapshots of the plume, while the right column shows the time-averaged plume snapshots over 60 timesteps, spanning a duration of 60 sequential minutes in total, under distinct background wind speeds but with a constant flux rate. Based on this simulation, we see that the plume varies rapidly in shape and orientation from snapshot to snapshot due to turbulence. The temporal averages in the right column also still exhibit some structure as we only averaged 60 individual snapshots. Overall, the simulated plumes from the LES closely resemble actual plumes from remotely-sensed observation as shown in Fig. 1. The instantaneous plumes exhibit non-Gaussian behavior; sometimes the plume can even be discontinuous as eddies can rupture the plume structure. However, we found that the total enhancement across the scene (the IME) remains rather constant over time for a given wind speed and flux rate, making it a reliable variable for performing the flux inversion of the source. In addition, we also found that the plumes have distinct features in both magnitude and spatial characteristics for different wind speeds, which are evident in the plume snapshots as well as their ensemble means shown in Fig. 4.

Fig. 5 illustrates the differences between the synthetic measurements for AVIRIS-NG and HyTES over the same plume for three different wind speed conditions. Because the column averaging kernel of the HyTES is close to zero near the ground, the synthetic measurements for HyTES miss parts of the plume near the surface, and detect only the parts of the plume that have risen high enough. This is consistent with the averaging kernels shown in Fig. 2. This is especially apparent for the case of high wind speed where the majority of the $CH_4$ is advected horizontally resulting in a plume remaining near the ground. The result in Fig. 5 is in accord with the comparison between the observed AVIRIS-NG and HyTES scenes in Fig.1 during the first overpass. This potentially indicates that the plume at this time remains mostly near the ground, which may not always happen in the same way for the coal mine venting shaft, which is emitting above the ground surface. The insensitivity of HyTES near the ground makes it complicated to locate the source accurately and there are additional uncertainties in the methane retrievals associated with averaging kernels that vary with environmental conditions (Kuai et al., 2016). The advantage of the HyTES instrument, on the other hand, is the fact that in principle it can operate at night when there is no sunlight, which is a prerequisite for AVIRIS-NG instrument. For AVIRIS-NG, the total column $CH_4$ enhancement in each pixel is also better constrained given the averaging kernel is approximately one throughout the column. For these reasons, we proceed to focus only on AVIRIS-NG results in the current study, while we will study the information content of joint measurements in the future.

Multiple LES runs from a combination of typical point-source flux rates and wind speeds enable us to quantify the relationship between the actual source rate and the resulting IME for a given wind speed. This gives us the first step to invert the flux rate. Furthermore, we show how different wind speeds affect this relationship for the flux inversion. The output from the LES gives us not only the IME but also the spatial distribution of the plume snapshots that correspond to a given pair of flux rate and wind speed. We analyze how the morphology of the plumes is linked with the underlying background wind

speeds. This helps us understand how we can use the remotely-sensed airborne imagery of the plume to predict the wind, and thus ultimately the flux rate, together with its associated errors.

In our analysis, we primarily refer to the wind speed in each scene from our model runs by using the geostrophic wind speed, as opposed to the instantaneous wind at 2 m (U-2) or 10 m (U-10) above ground which is usually used in literature. For reference, the average U-10 across the horizontal domain in our run ranges approximately from 0.4 to 0.7 of the background geostrophic wind speed in the run. The main reason is that our output snapshots from each LES run is written out every minute, thus we only have the information of the U-10 and the plume structure at every minute, which can change rapidly in direction and magnitude. However, the overall structure of the plume at any given instance could be influenced by the average wind cumulatively from the past minute. The constraint on the output that we have makes it ambiguous to choose what values of near surface winds should be applied when making the prediction of the flux rate from the spatial structure of a plume snapshot. We thus resort to using a background wind speed, which, in turn, is one of the key governing drivers for U-10 itself. While using the large-scale background wind speed might not be as accurate as the ideal case of having continuous U-10 output, it provides a robust correlation with the overall pattern of the plume (see Section 6.2). In other words, in the following, we are using the shape of the plume to predict the value of background geostrophic wind speed that underlies the wind that has driven $CH_4$ from the point source into the detected plume over that geographical location, and use that background wind speed to quantify the source rate.

## 6.1 Source Flux Rate and the IME

For each wind speed and flux rate, we have 60 snapshots of methane plumes from the LES model output, with a temporal interval of one minute. We can thus directly compute the mean and the standard deviation of the IME across these snapshots. Although the shape of a plume can vary strongly in time, the IME is relatively stable, varying only within approximately 20% among snapshots under the same wind speed and flux rate. This emphasizes the benefit of using the IME to characterize methane in the scene because the total sum of the gas in the scene remains approximately the same regardless of the advection of methane from one pixel to another with time. This can potentially induce less uncertainty compared to other mass balance approaches where the measurements are commonly location-dependent. The mean values corresponding to various background wind speeds and flux rates are plotted in Fig. 6. The uncertainties reflect the standard deviations of the IME within all 60 temporal snapshots.

The plot of the IME and flux rate at different wind speeds reveals two noticeable findings: as expected, there is a significant dependence of the relationship between the IME and flux rate on wind speed but there is also a non-linearity, which has been ignored in previous studies. The non-linearity can be explained from the fact that we impose a detection threshold to mask out the plume. In the absence of a detection threshold, the scaling between flux rate and IME would be perfectly linear, as was assumed in Frankenberg et al., (2016). However, as the fraction of pixels with methane enhancement below the detection threshold varies with flux rate and wind speed, the truncated IME below the threshold can induce a considerable non-linearity.

The stronger the flux rate, the higher the number of pixels above the threshold used to calculate the IME. Fig. 7 illustrates this connection by showing the percentage of the total enhancement that is missed because of specific thresholds. We use three different flux rates (90, 180, 360 kg h$^{-1}$) to illustrate the non-linearity. We can see that when the flux rate drops by a factor of 2, the missing amount does not necessarily decrease by the same factor. How the IME is scaled up with the flux rate depends on the spatial distribution of the plume: if the methane is concentrated in a small area, then it is more likely that a stronger flux rate will make the column enhancements exceed the threshold, as opposed to when the plume is more dispersed, in which case some pixel enhancements will be too diluted to be detected even at a strong flux rate. This is the primary reason why the IME varies with the flux rate with different degree of non-linearity at different wind speeds as found in Fig. 6. The background wind speed is the integral component that drives the spatial distribution of the plume and correlates the IME with the flux rate. This means that in order to achieve a reliable flux inversion, both the IME and the effective wind speed over the scene of the point source must be known.

The key question in our study is can we predict the underlying background wind speed associated with the observed plume by its spatial characteristics rather than relying on ground measurements or reanalysis data. This is investigated in the following section.

## 6.2 Wind Speed and Plume Morphology

As can be seen in Fig. 4, the spatial distribution of the plumes varies under different wind speeds. Visually, the shape of simulated CH$_4$ plumes provides qualitative intuition on the origin, wind direction and relative strength of the background windspeed. At a higher wind speed, plumes tend to be more elongated, whereas at a lower wind speed, plumes tend to be more spread out around the origin. We quantify the characteristics of the plume by first constructing an angular mass distribution for each snapshot: we count the mass within the angular bin size of 0.5° sweeping across the scene with the center at the origin. We then find the angle at which the mass of methane splits into a 50% ratio and define that as the main axis of that plume snapshot. The plume snapshot is then rotated such that its main axis aligns with the x-coordinate. We can then plot the angular distribution across the plume as well as the Cartesian distribution along the plume, as illustrated in Fig. 8, for every single snapshot. This procedure allows us to find the ensemble-averaged plume distributions for a particular wind speed where the ensemble members consist of the rotated snapshots from all available time outputs in the model runs at various flux rates in the range of our interest, 10–1000 kg h$^{-1}$.

Fig. 9 shows that the angular distributions of the plume can be distinguishable under different wind speeds. Evidently, the angular distribution of the plume at highest wind speed of 10 m s$^{-1}$ is narrower than the rest on average, and the angular spreading becomes increasingly wider for lower wind speeds. Motivated by this finding based on the average distribution, we quantified the relationship between the angular spreading of the plume and the wind speed. For each snapshot, we calculated the cone width of the plume defined as the angles between the 10$^{th}$ and the 90$^{th}$ percentiles from its angular mass distribution. The mean and the standard deviation of the cone width corresponding to a given wind speed were then computed from an ensemble of 60 temporal snapshots and various flux rates. The result of this analysis is plotted in Fig. 10 and shows a

monotonically decreasing cone width with respect to wind speed. Our choice of parameterization in Fig. 10 is an exponential fit, which adequately captures the present relationship without overfitting. This result illustrates that the cone width is a metric that can differentiate wind speeds based on using only the spatial distribution of the plume. This finding, together with the variation of IME with flux rate (Fig. 6), can therefore provide flux inversion without the need for ground measurements. The next section describes steps for estimating the flux rates and its associated uncertainties.

## 6.3 Flux Inversion and Error Analysis

Based on the IME and plume morphology of any given scene, we can estimate the flux rate. First, according to Fig. 6, for a given value of the IME observed in the scene, we can find what are possible range of fluxes for each wind speed from the lower and upper estimate of 1 standard deviation. We can then parameterize this relationship between the flux rate and the wind speed for this particular value of the IME. An example for the case of the observed IME of 50 kg is demonstrated in Fig. 11. Secondly, based on the spatial distribution of the plume in the scene, we can follow the procedure to construct the angular mass distribution. Based on Fig. 10, using an angular width measured from the plume, we can predict the wind speed from the fitted curve. The associated uncertainties of the wind speed are approximated by the lower and upper estimate of 1 standard deviation. We assume that by projecting a value of plume width onto the corresponding range of wind speeds within 1 standard deviation range, we obtain uncertainties for predicted wind speed that approximately represent 1 standard deviation error for the wind speed distribution. The wind speed and its uncertainty can hence be translated into the estimate of the mean flux rate as well as the corresponding uncertainties from the relationship of the flux rate and wind speed, as in Fig. 11.

With this approach, we selected 90 random snapshots with random prescribed flux rates and wind speeds. We predict the flux rate from the IME and the spatial distribution of each of plume scene and compare it to its actual prescribed value, as shown in Fig. 12. The average of the percentage differences (in absolute terms) between the predicted value and the actual value for single point source predictions is approximately 30%. The $\chi^2$ value from the predictions in Fig. 12 is 3.84 suggesting that the error variance may tend to be slightly underestimated for an individual point source prediction.

Nevertheless, the results shown in Fig. 12 demonstrates that this method permits estimation of total emission flux rate. Most importantly, accounting for non-linearities and variable wind speed helps to avoid systematic biases. Thus, the method employed here can minimize systematic errors that could be induced by assumptions on wind speed. To verify this point, we performed an aggregation analysis by bootstrapping 30 plumes out of 500 plumes of various flux rates and wind speeds, with 3000 repetitions. The sample size of 30 is chosen arbitrarily but is large enough to represent a situation for the estimation of total fluxes from a region. The comparison between the predicted and the actual total flux aggregated over 30 plumes is shown in Fig. 13. The predictions lie close to the actual aggregated fluxes, as demonstrated by the concentration of points near the 1-to-1 line in Fig. 13, implying that there are no significant systematic biases in our method. The mean of absolute differences from all these aggregates is 5.1% with a standard deviation of 3.9%, while the average of all differences (negative and positive) results in 2.9% with the standard deviation of 5.9%.

To further demonstrate the validity of this method, we applied it to a controlled release experiment from a natural gas pipeline located at Victorville, CA (34.8°, -117.3°) on October 11$^{th}$, 2017 with a flux rate of $89 \pm 4$ kg h$^{-1}$. Based on a sample of the actual AVIRIS-NG scene over the source location (Fig. 15), we calculated the IME and constructed the angular distribution of the plume to obtain its width to deduce the wind speed. The geostrophic wind speed is predicted to be $3.3 \pm 1.2$ m s$^{-1}$, compared to the surface sonic wind at the source measured at 1.6 m s$^{-1}$. This is consistent given that geostrophic wind is typically about $1.4 - 2.5$ times higher than the surface wind speed in the LES output. We used this deduced wind speed to predict the flux rate and its associated error as described at the beginning of this section. The value that we predict is $118 \pm 30$ kg h$^{-1}$, consistent with the actual release flux within the error estimate.

Furthermore, we applied our method to multiple overflight AVIRIS-NG scenes from Fig.1. The fitted flux rates are within a consistent range: 1275, 1033, 1397, and 926 kg h$^{-1}$ respectively. The mean of these estimates is thus 1158 kg h$^{-1}$ and the standard deviation is 187 kg h$^{-1}$.

## 6.4 Sensitivity Analysis for Different Heat Fluxes

In our LES simulations for this study, we primarily set the sensible and latent heat fluxes to the typical condition during the Four Corner field campaign. Changing the condition of these surface heat fluxes can potentially affect the vertical structure of the simulated plumes and the dynamics of the plumes in time. Nevertheless, our method involves the column-integrated enhancement and hence is not significantly impacted by the surface heat fluxes. To verify this point, we performed sensitivity analysis by running additional LES experiments with different combination of sensible and latent heat fluxes (SH and LH respectively): (1) SH = LH (220 W m$^{-2}$) and (2) SH (200 W m$^{-2}$) < LH (400 W m$^{-2}$). These two additional scenarios contrast with the typical condition that was previously used i.e. SH (400 W m$^{-2}$) > LH (40 W m$^{-2}$), and cover a common range of surface heat flux conditions. The background wind speed is kept the same as 4 m s$^{-1}$. The result from our runs are demonstrated in Fig.15, where the relationship between the IME and flux rate is found to be approximately the same, remaining within 1 standard deviation error from the original scenario in the previous analyses. This implies that the uncertainties associated with the change in these conditions will not significantly impact our method and are captured well with the range of errors we have analyzed.

## 7 Discussion and Conclusion

In this study, we showed that Gaussian plume modeling cannot be used for a meaningful comparison with observed methane plumes from a point source. Thus, Large-Eddy Simulations (LES) were used to generate realistic synthetic measurements of methane plumes under different background wind speeds and source flux rates. This allowed a comparison of the performances of two considered instruments, one measuring in the short-wave infrared (AVIRIS-NG) and the other in the thermal infrared (HyTES), resulting in widely different vertical sensitivities towards methane enhancements. The AVIRIS-NG was found to provide an unambiguous identification and quantification of the methane source as it is sensitive to methane throughout the air column. While the HyTES instrument has the potential for night time observations, variations in the

integrated methane enhancements depended highly on vertical plume structure, rendering interpretation more challenging. While we attempt to make use of the vertical information in the future, we focus this study on results from the AVIRIS-NG synthetic plume measurements. Using the IME method and a large ensemble, we derived the relationship between the detected IME of a plume and its source flux rate. This relationship is found to be non-linear because of the device detection threshold, which causes a variable fraction of the true IME to fall below the detection limit. In addition, the inversion of IME to an accurate flux rate depends strongly on the wind speeds during the measurements. This finding is expected and confirms the significance of wind speeds on the methane point-source flux estimations from remote sensing data. To study whether we can gain additional information from the plume shape itself, we performed an analysis on a large ensemble of plume snapshots from wide-ranging source flux rates and wind speeds. We found that the angular width of the plume negatively correlates with the wind speed, allowing us to constrain the effective wind-speed from the shape itself. The angular width is defined based on the plume angular distribution around its main axis and is found to be effectively independent of the source rates.

Using the relationship between the IME and the flux rates for different wind speeds together with the connection between plume shape and the wind speed, we can disentangle the source flux rate based on an observed snapshot of the plume which provides both the IME and the spatial distribution. Our error analysis of this method applied on randomly generated snapshots of various flux rates in the range of 10-1000 kg h$^{-1}$ showed an error of around 30% on average for an individual point source estimate. Given that point sources are highly uncertain and also fluctuate in time, this single measurement error appears acceptable. More important than single measurement precision is accuracy for larger ensemble averages, which informs regional emission estimates. Thus, we also performed an error analysis for aggregated flux estimates from 30 plumes. We used bootstrap sampling and found the aggregation error estimate to be in the range of less than 10%. This provides a significant improvement from other pre-existing approaches that rely on wind data, for which reliable meteorological reanalysis data might not be available at high spatial resolution everywhere.

Furthermore, our method is validated by the application of this method on an actual scene from a controlled release experiment from a natural gas pipeline in 2017, which demonstrated an error of 32% from the controlled flux rate of 89 kg h$^{-1}$, a notable accuracy given the simplicity of our algorithm that does not require wind speed data. This provides added value in quantifying methane point source emissions especially in locations where atmospheric reanalysis products and surface meteorological observations are not available.

It should be noted that altering the device detection threshold level in our synthetic modelling to higher values does impact the robustness of the correlation between the plume width and the wind speed. In this study, we set the threshold to 500 ppm-m to match the capabilities of the current instrumentations. Future instruments with improved gas sensitivity (Thorpe et al., 2016b), will likely improve our ability to estimate emission rates. Repeat overflights that result in multiple snapshots of the same source can also further reduce uncertainties from transient variations of the plume due to turbulence. Another aspect is that our current LES does not yet model direct emission that could be released at height above the ground. Incorporating this feature into our future analysis may provide even more realistic methane plume simulations. Despite these limitations, this current study is a first step proving the potential of the method.

In this study, we have demonstrated the ability to estimate flux rates of methane point sources based solely on the remotely sensed column methane enhancement without the need for ground measurements or weather reanalysis data. This method could be applied to recent large-scale flight campaigns to improve previous emission rate estimates. This also has immediate implications for future AVIRIS-NG flight campaigns, in particular over parts of the world lacking available wind data. The methodology described in this study could also be applied to anticipated satellites that will provide methane measurements at finer spatial resolutions than currently available. A path towards an improved understanding of the regional methane budget as well as insights into methane source distributions by categories is made possible.

**Acknowledgements**

This work is part of SJ's NASA Earth and Space Science Fellowship (NESSF). We acknowledge the Resnick Sustainability Institute at Caltech for their kind support with computing resources. This work was supported in part by NASA's Carbon Monitoring System (CMS) Prototype Methane Monitoring System for California. We also thank NASA's Earth Science Division, particularly Dr. Jack Kaye, for continued support of AVIRIS-NG and HyTES methane science. Additional funding was provided to JPL by the California Air Resources Board under ARB-NASA Agreement 15RD028 Space Act Agreement 82-19863 and the California Energy Commission under CEC-500-15-004. A portion of this research was carried out at the Jet Propulsion Laboratory, California Institute of Technology, under contract with the National Aeronautics and Space Administration (NNN12AA01C). We thank the AVIRIS-NG team and colleagues at Pacific Gas and Electric Company for their support for controlled release experiments.

**Code/Data Availability**

AVIRIS-NG data are publicly available via https://aviris-ng.jpl.nasa.gov/benchmark_methane_data.html, HyTES L2 and L3 data are available for ordering free of charge at http://hytes.jpl.nasa.gov/order

**Author Contribution**

SJ performed the analysis and wrote the paper, with the overall research objectives advised by CF. GM ran the LES model, provided output and guided the analysis, AT, RD, EK and DT provided AVIRIS-NG and HyTES datasets and supported writing and data analysis.

**Competing Interests**

None

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

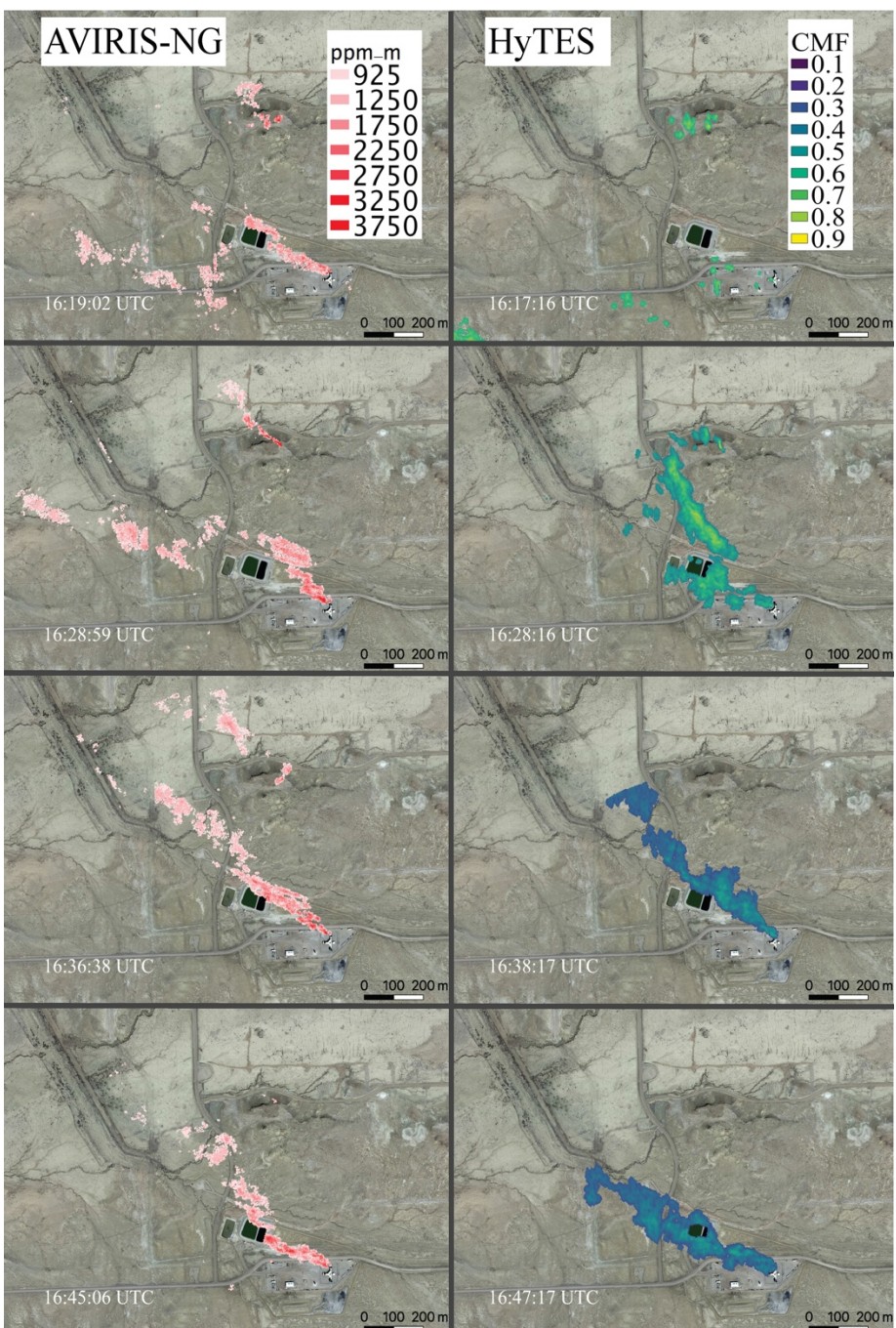

**Figure 1: Methane plume over a venting shaft in the Four Corners region, observed from four individual airborne instrument AVIRIS-NG overpasses (2.8 m spatial resolution) at 7-9 minutes apart on April 22[nd], 2015 between 16:19:02 and 16:45:06 UTC (left) compared with observations from HyTES overpasses (2.3 m spatial resolution) in the similar interval between 16:17:16 and 16:47:17 UTC (right). The background is from Google Earth imagery.**

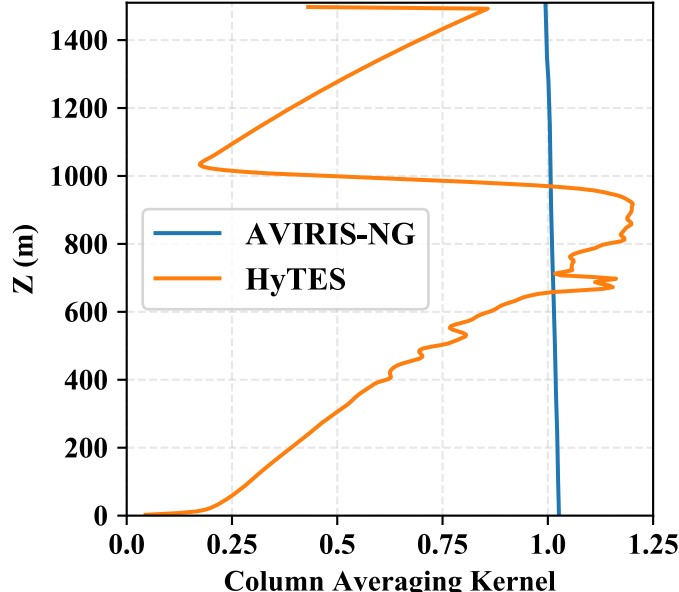

**Figure 2: Column averaging kernels for two instruments, AVIRIS-NG (in blue) and HyTES (in orange), as a function of height. The altitude on z-axis is given above ground level. In the Thermal case (HyTES) the flight altitude is an important factor for the CAK. The CAK of HyTES was computed for an altitude of about 3 km. For the shortwave range, however, the CAK of AVIRIS-NG is not impacted significantly by flight altitude.**

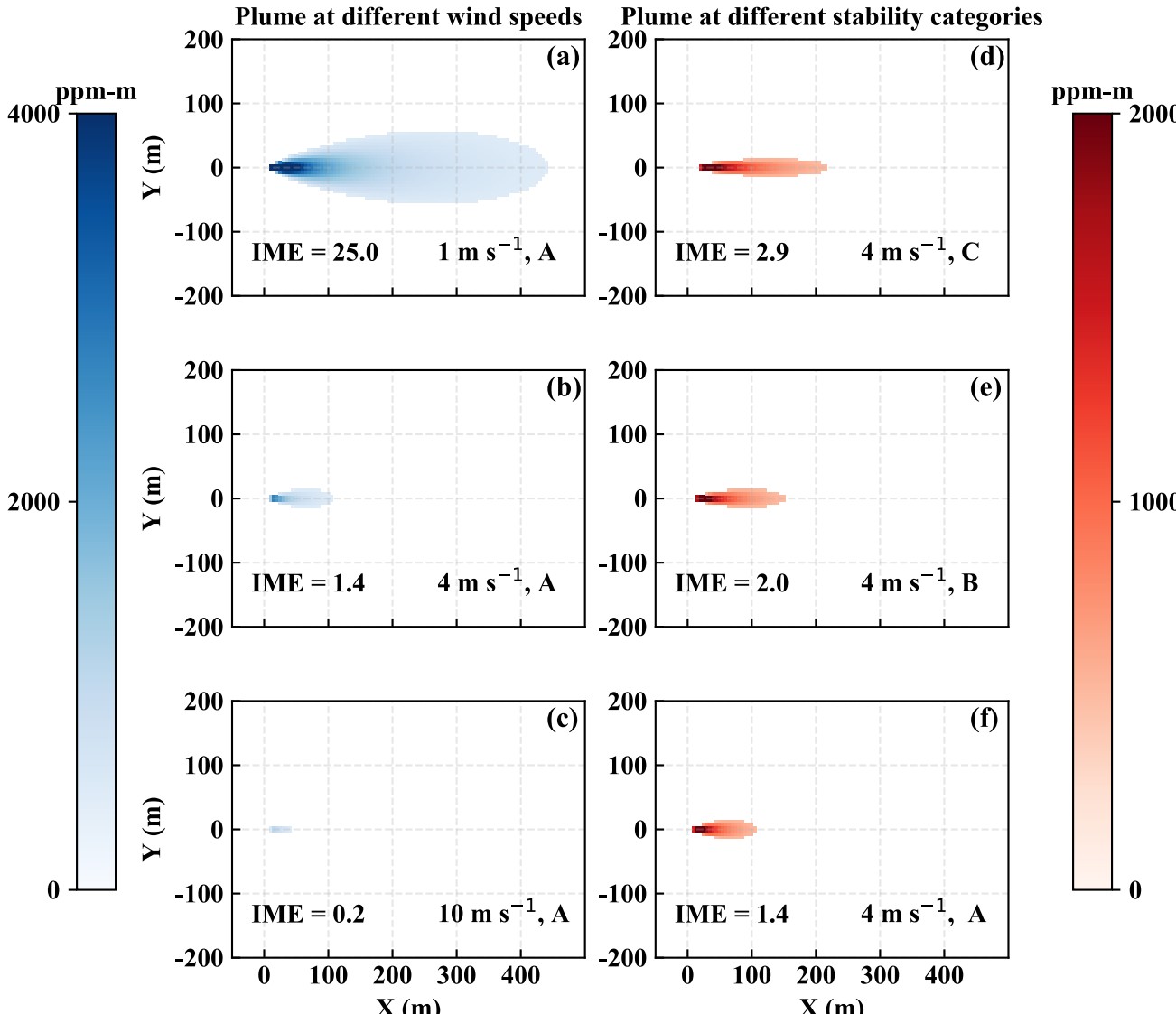

**Figure 3: (a) – (c) Gaussian plumes under wind speeds of 1,4, and 10 m s$^{-1}$ respectively, with Pasquil stability type "A= very unstable". (d) – (f) Gaussian plumes under wind speed of 4 m s$^{-1}$ in the stability type "A=very unstable", "B=unstable", and "C=slightly unstable" respectively. All cases are with flux rate of 300 kg h$^{-1}$ and detection threshold set to 500 ppm-m. The IME is calculated over the entire scene and is in kg. The wind speed shown in this Gaussian model is at plume levels.**

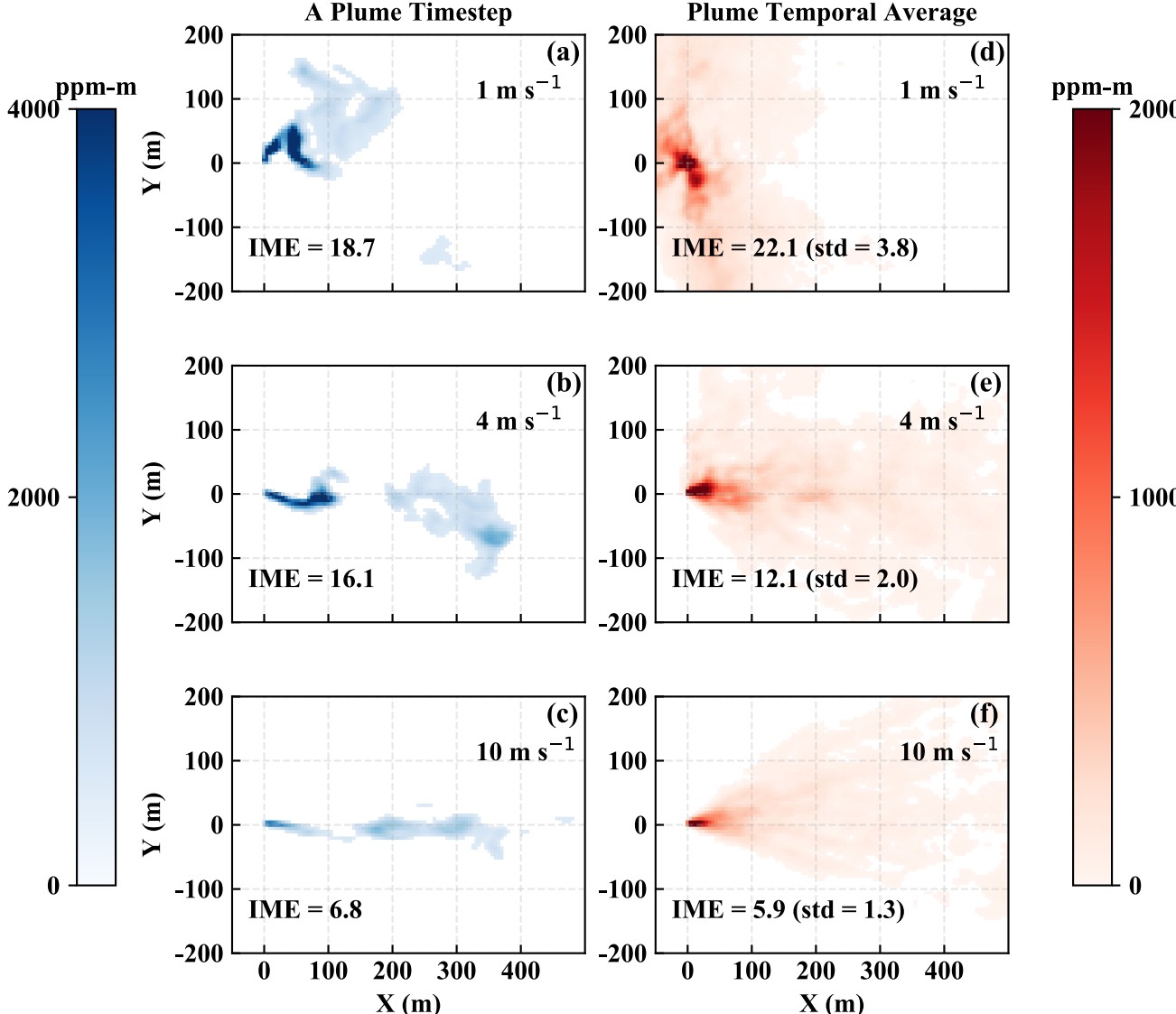

**Figure 4: (a) – (c) Snapshots of simulated plumes under wind speeds of 1,4, and 10 m s⁻¹ respectively. (d) – (f) time-averaged plumes from 60 timesteps under the geostrophic wind speeds of 1,4, and 10 m s⁻¹ respectively. All with flux rate of 300 kg h⁻¹ and detection threshold set to 500 ppm-m. All are based on AVIRIS-NG averaging kernels. The IME is calculated over the entire scene and is in kg. Note that the temporal averages do not reach a true ensemble average as sample size are finite (i.e. the average still exhibit fine structure).**

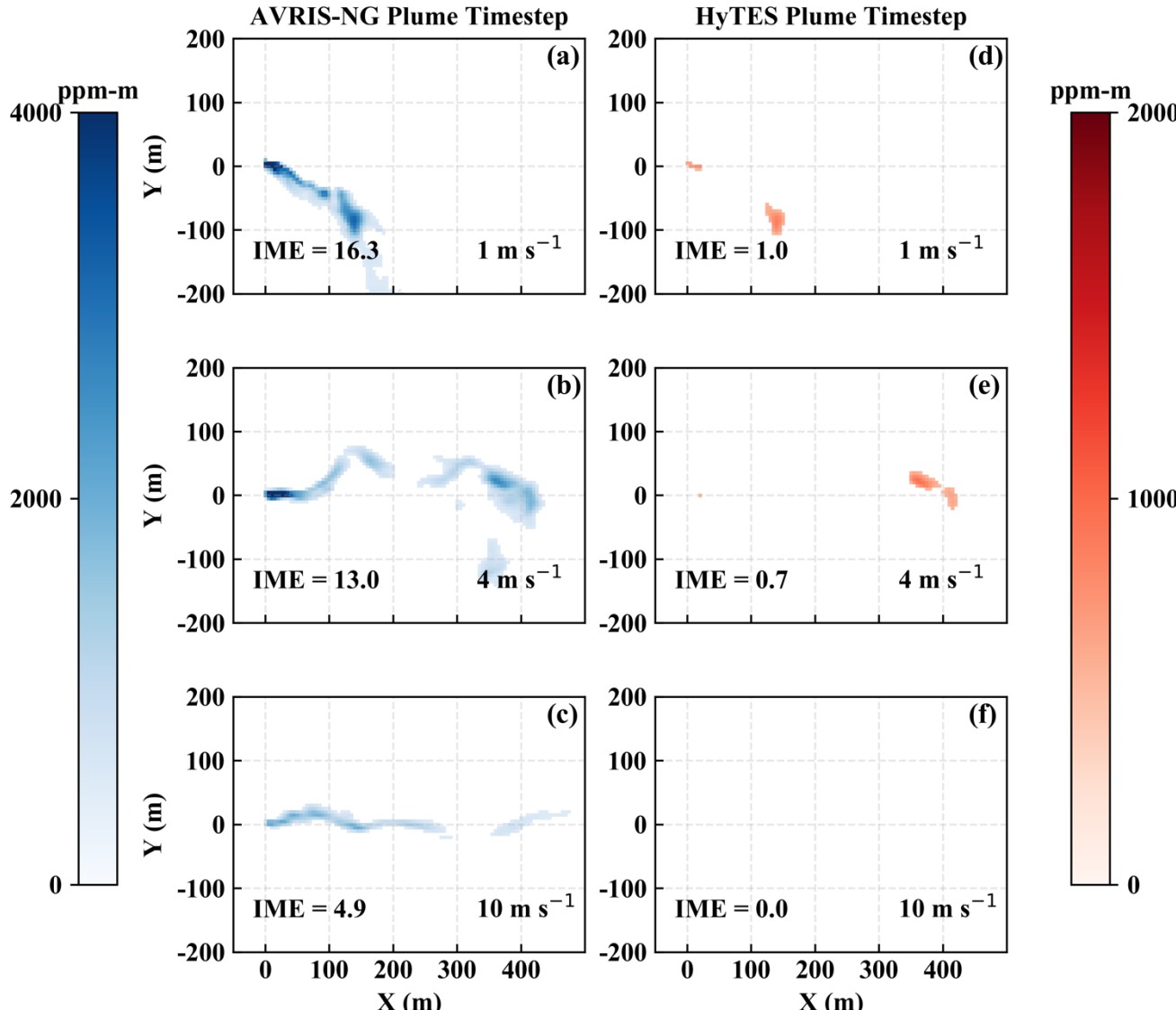

**Figure 5: (a)-(c) Snapshots from simulated plumes under 1, 4, and 10 m s$^{-1}$ respectively, when applying the AVIRIS-NG instrument column averaging kernel. (d)-(f) Snapshots from the exact same plumes as in (a)-(c) respectively, but applying the HyTES averaging kernel. The flux rates are all 300 kg h$^{-1}$ and the detection threshold is set to 500 ppm-m. The IME is in kg.**

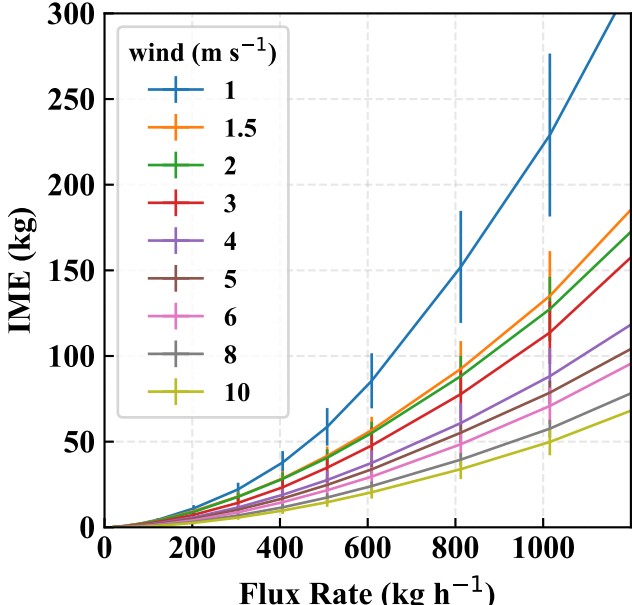

**Figure 6: Mean and standard deviation of the IME associated with a range of flux rates under various background wind speeds from 1 to 10 m s$^{-1}$. The detection threshold is 500 ppm-m.**

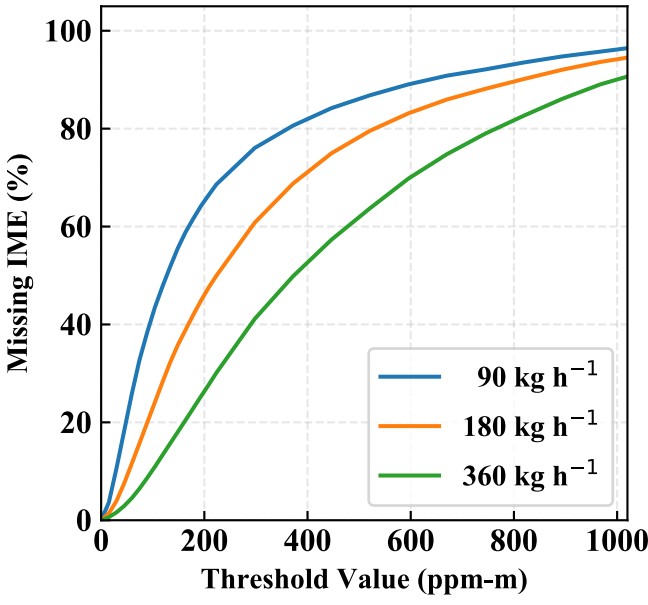

**Figure 7: Missing IME, shown as a percentage, for different ppm-m threshold values. Each curve corresponds to a prescribed source flux rate. The flux rates are incremented by a factor of 2.**

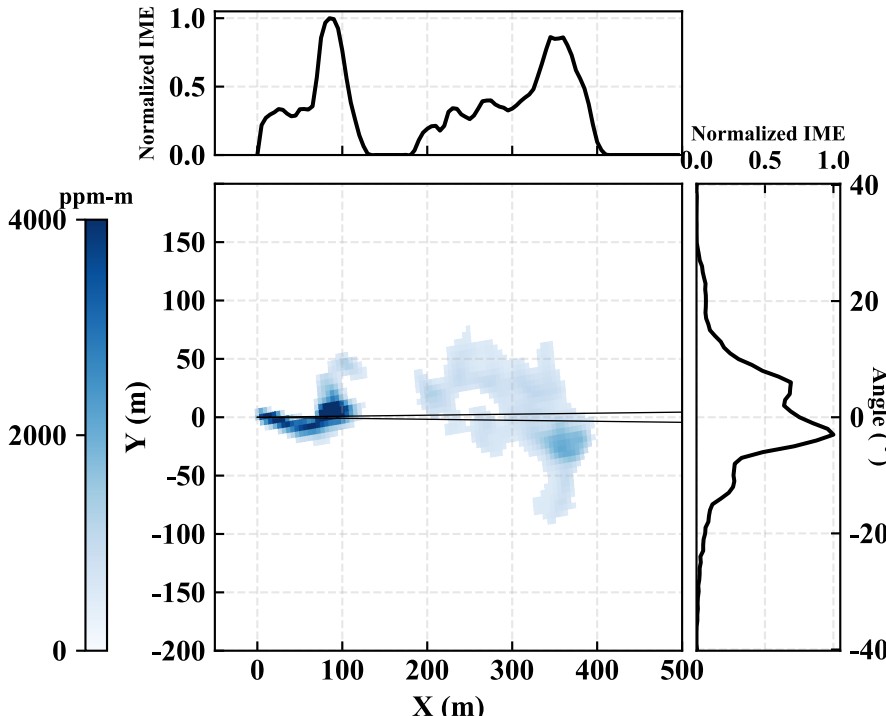

**Figure 8: A rotated plume snapshot from a run of 4 m s⁻¹ background wind speed and 300 kg h⁻¹ flux rate with its angular distribution of IME across the plume (right) and its Cartesian distribution of IME along the plume (top). The two black lines denote an angular bin of 0.5 degree that sweeps through the 2D plume to construct the angular distribution.**

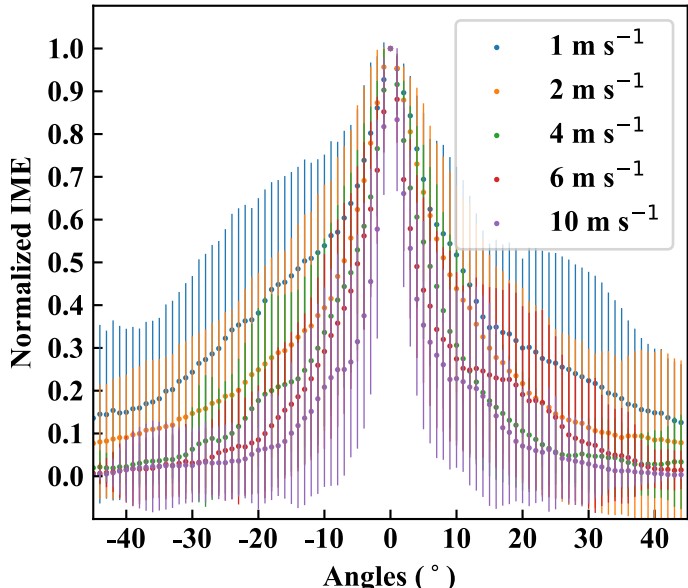

**Figure 9: Ensemble-averaged angular distributions of the plume, averaging over all available timesteps at various flux rates. Different colors represent different wind speeds. Each distribution is normalized by its maximum value. The vertical bars represent one standard deviation of the normalized IME at a given angle across all snapshots.**

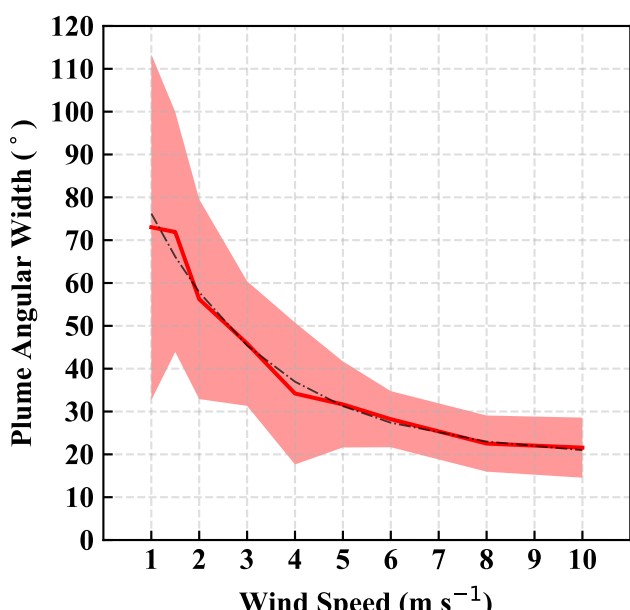

**Figure 10: Relationship between the wind speed and the associated cone width averaged over snapshots and flux rates. The dotted black curve represents the best fit by an exponential function. The shaded area represents one standard deviation from the mean plume angular width for each wind speed.**

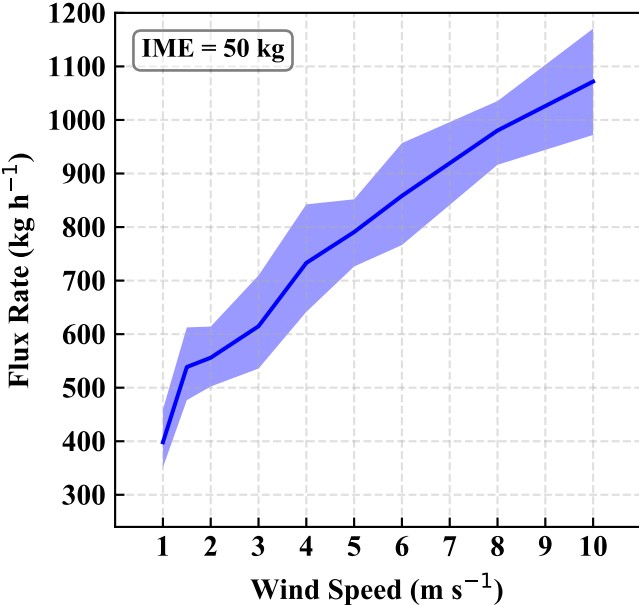

**Figure 11: Relationship between flux rate and wind speed for 50 kg IME. The shaded area represents one standard deviation from the mean flux rate at each given wind speed.**

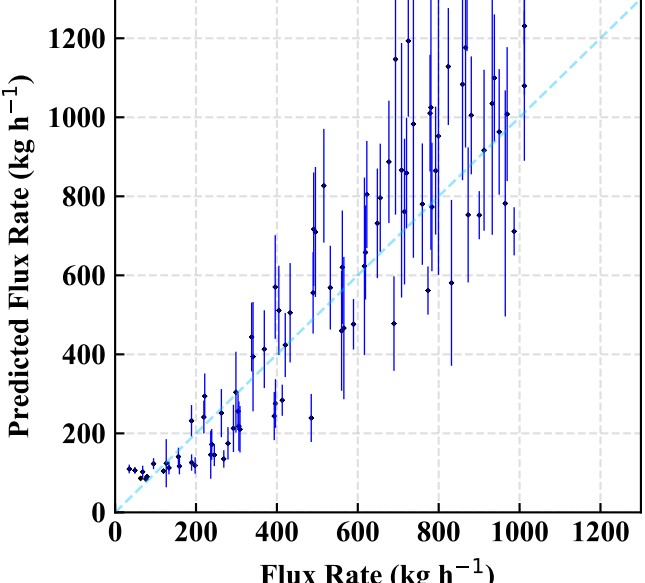

**Figure 12: Comparison between the prescribed flux rate in the model run and the predicted flux rate based on our method of using the IME and the angular width of plume in a given scene. The error bar represents uncertainties associated with the prediction of an individual point souce.**

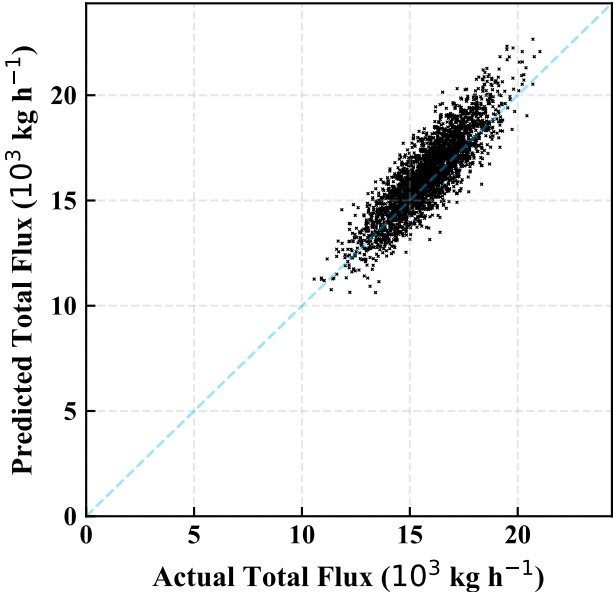

**Figure 13: Comparison between the predicted and the actual total flux of 30 plumes, from 3000 bootstrap rounds.**

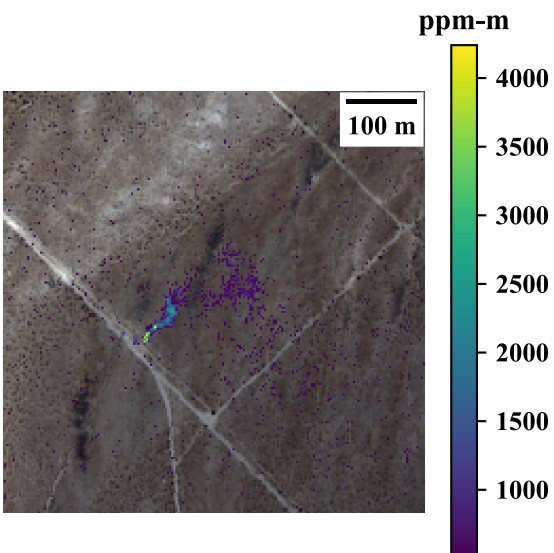

**Figure 14: An observed AVRIS-NG scene in a controlled release experiment from a natural gas pipeline located at Victorville, CA (34.8°, -117.3°) on October 11th, 2017 with the flux rate of $89 \pm 4$ kg h$^{-1}$.**

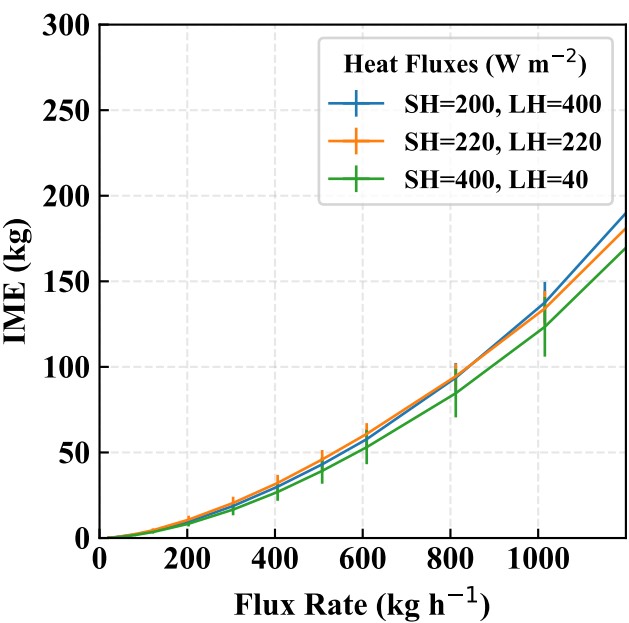

5  **Figure 15: Relationship between IME and flux rate under different sensible and latent heat fluxes of 200 and 400 W m$^{-2}$ (blue), and 220 and 220 W m$^{-2}$ (orange), compared to the original simulation sensible and latent heat fluxes of 400 and 40 W m$^{-2}$ (green). All cases are under the wind speed of 4 m s$^{-1}$. The detection threshold was 500 ppm-m.**