# Peer review of "Towards accurate methane point-source quantification from highresolution 2D plume imagery"

_Atmospheric Measurement Techniques, 2019_

## Referee Comment (RC1) · Anonymous Referee #1 · 15 Jun 2019

**General comments**

The publication "Towards accurate methane point-source quantification from high-resolution 2D plume imagery" by Jongaramrungrang et al. deals with the important aspect of inverting atmospheric concentration gradients to fluxes or emission rates of not well-constraint CH4 point sources. In order to invert these observations, wind information in the measured area is needed and equally important as the concentration measurements themselves. However, as outlined in the introduction of the manuscript, acquiring wind information, preferably, simultaneously and at an adequate spatial and temporal resolution can be a challenging task. Therefore, the authors propose a new method, which solely relies on spatially resolved imaging data from airborne remote sensing instruments and high resolution large eddy simulations (LES) to estimate the prevailing wind speed during the time of the overflight. The wind speed is then derived from the shape of the observed plume and used during the inversion process to compute a flux of the investigated sources.

The described method is a novel and promising approach to quantify CH4 point source emissions from aircraft but also potentially from space without having to rely on real wind measurements. The manuscript fits well in the scope of AMT and I recommend publication after some modifications along the line of the comments below.

In general, the manuscript is well written. The method is described in a comprehensible way, however, some additional information concerning the figures would improve their readability (see also specific comments). Additionally, the authors should add some more information regarding the stated errors and their propagation to the predicted fluxes (see also specific comments).

In my opinion, the manuscript could be strengthened by adding more extensive comparisons and applications to real data. So far, most parts of the approach were developed based on (theoretical) model (LES) studies, whereas only few real observations were used to support the novel approach. Therefore, I would recommend to expand on the already given examples in the manuscript: (1) controlled release experiment and (2) analyses of overflights shown in Figure 1 (see also specific comments).

**Specific comments**

**P2, L12:** Consider to also add publications regarding the HyTES instrument (e.g., Hulley et al., 2016) or other imaging instruments, which also have CH4 point sources successfully detected, e.g., MAKO (Tratt et al., 2014).

**P2, L24f:** I would suggest to either only focus on remote sensing studies (by satellite and aircraft) and remove Conley et al., 2016, or to better distinguish between remote sensing and in-situ studies. If in-situ studies are included I would also recommend to add, e.g., Cambaliza et al., 2015, Gordon et al., 2015, and Lavoie et al., 2015., which have also performed extensive analyses regarding airborne in-situ observations and resulting fluxes.

**P3, L6:** "… under various background wind speeds and **surface heat fluxes**.": Later in the manuscript (P6, L14), it is stated that only one value for the heat fluxes was used ("The surface sensible and latent heat fluxes are 400 and 40 W/m².").

**P3, L11:** "… and presence of ancillary information on the **actual wind speed (Section 5.3)**.": Based on this statement, I would expect to see a comparison between the derived wind speed based on the new method and actual wind observations or reanalysis data in Section 5.3., however, I was not able to find a paragraph referring to actual wind speed observations or reanalysis data.

**P4, L1:** Why not also showing a real example observation of HyTES. Maybe, there is one available for the same scene as shown in Figure 1. If this is the wrong place, because in Figure 1, the authors intend to show the high variability of the plume structure during multiple overflights, I would recommend, if available, to add a comparison of an AVIRIS-NG and HyTES observation after Figure 5, where simulated plumes observed by the two instruments are compared.

**P4, L1ff:** This would also be a good place to add some more information regarding the applied retrieval algorithms. As Frankenberg et al., 2016, are already cited multiple times and the plume shown in Figure 1 (bottom right) appears to be similar to the one in Figure S1 in Frankenberg et al., 2016, I assume that either the matched filter technique or the IMAP-DOAS method, described in that publication, has been used to retrieve the columns shown in Figure 1 or the ones from the controlled release experiment at the end of Section 5.3 used for a flux inversion.
Similar for the HyTES algorithm if real observations are added.

**P4, L26:** What is the basis of the chosen threshold of 500 ppm-m? As stated in (P3, L30), it is connected to the measurement precision of the AVIRIS-NG instrument and is thus a property of the instrument (Why does it then differ from the value of 200 ppm-m applied in Frankenberg et al., 2016, Section 'IME' in 'Materials and Methods'?).
Furthermore, what is the reasoning in using the same threshold also for HyTES simulations, e.g., as shown in Figure 5?

**P5, L5:** Please add some explanation for the 'sum' and the parameter 'm' in Equation 2 to the text.

**P6, L10-15:** What is a reasonable range of the proposed values (initial inversion height, potential temperature, specific humidity, surface sensible and latent heat flux) for initializing the LES model? Are these educated guesses or are they based on actual field or reanalysis data?
Given these ranges, the authors could also verify their assumption "… our method … should not be significantly impacted …" in the conclusions (P12, L6) by performing various LES runs using different initial values. I would agree with the authors that the column-integrated enhancements are not significantly influenced by the surface heat fluxes if the threshold were 0 ppm-m. However, I am curious if and how Figure 6 and 7 might change under various initial conditions, or whether they are well within the error bars.

**P8, L27:** Do the authors only mean Frankenberg et al., 2016, by "… has been ignored in previous studies." or are there further ones?

**P9, L33:** Could the authors add the fitted polynomial also to Figure 10? Why do the authors use a fifth-degree polynomial? Is there a physical relationship, which relates wind speed and plume angular width by a fifth-degree polynomial or is it just the 'best' fit?
Could the relationship in Figure 10 also be explained by an exponential curve? Assumption: The relation in Figure 10 is determined by averaging over an ensemble of LES realizations. If this is done, I expect the resulting plume to be approximately Gaussian as seen in Figure 4, e-f.

**P10, L5-13:** Basically, in this paragraph the authors summarize their developed method. I have some questions/comments to the used example.
a) The error bars (shaded area) shown in Figure 11: Are they related to the errors shown in Figure 6 or to Figure 10?
b) Have the flux rates, shown in Figure 11, been corrected for the missing IME (as indicated in Figure 7)?
c) How is the error (1-SD of the plume angular width) shown in Figure 10 related/translated to the wind speed error, which then linearly propagates to the estimated flux?

**P10, L14-17:** Could the authors be more precise in terms of the given "average percentage error" of 30%? I assume it is the mean value of the vertical error bars shown in Figure 12. However, as in reality not only entire fields/regions of CH4 sources (as then investigated in P10, L18-25) are investigated but also single plumes, which are typically observed only once, an interesting measure would also be the average of the absolute differences (also in percent) of the predicted flux rates and the corresponding prescribed flux rates. This would give an idea of the magnitude of the bias one can expect from the method. I assume that the observed bias in the flux for the controlled release experiment of ~32% lies within this computed theoretical value.

**P10, L25:** What do the authors exactly mean by "mean percentage of error"? Is that the average of all differences between actual and predicted flux OR is it a similar error as computed for Figure 12 in P10, L14-17? If the authors refer to the latter one, I would suggest to also compute the average of all differences between actual and predicted flux (not the average of the absolute differences as in the previous comment). For example, a positive or negative value would then quantify an over- or underestimation caused by the method on average. The same exercise can be done for Figure 12 because it appears (as for Figure 13) that more predicted fluxes lie above the 1-to-1 line then below especially for larger fluxes.

**P10, L26ff:** The possibility to compare the novel approach, which is mostly based on 'theoretical' models, to real data is a huge strength of the publication. Therefore, I would recommend to expand this part of the publication. Some starting points are already given.
First of all, the authors could add some more information regarding the already analyzed **controlled release experiment** allowing for a better judgement by the reader. Useful information would be **(a)** a figure showing the overflight and the retrieved plume and CH4 column enhancements (similar to Figure 1), **(b)** the fitted wind speed, which is then used to invert the IME to a flux, and **(c)** as the observation is based on a controlled release experiment, do the authors have access to real wind observations on-site or at least to meteorological reanalysis data, which can then be compared to the fitted wind to test its plausibility?
Additionally, the authors nicely show multiple overflights of one source within ~25 minutes by the AVIRIS-NG instrument in **Figure 1**. It would be an interesting opportunity to apply the developed method to the four overflights shown in that figure and discuss the resulting fitted winds and inverted fluxes.

**P10, L29:** I assume the given estimated emission of 118 kg/hr is already corrected by the potentially missed IME as indicated in Figure 7, right? Additionally, could the authors elaborate on what error sources are included in the error estimate (of 30 kg/hr) of the predict flux.

**P15, Figure 2:** Please clarify whether altitude on z-axis is given in meters above sea level or above ground level. Additionally, please harmonize the minimum altitudes of the computed averaging kernels, either to 0 m or to a specific surface elevation. Consider also adding the aircraft altitude(s) which the examples CAKs are valid / have been computed fo.

**P16, Figure 3:** Consider adding the true IME (idealized threshold of 0 ppm-m) to the caption. Additionally, consider adding labels for the stability classes to the caption so that the reader sees immediately their meaning without looking up the relevant information in the cited publications, e.g., A = very unstable; B = moderately unstable; ...

**P17, Figure 4:** Consider adding the true IME (idealized threshold of 0 ppm-m) to the caption (as suggested for Figure 3), and the variance of IME (of the 60 individual snapshots) to each plot in the

right column for the three cases of wind speed so that the reader can assess the statement from (P7, L10f).

**P18, Figure 5:** Consider adding the true IME (idealized threshold of 0 ppm-m) to the caption (as suggested for Figure 3)

**Figure 3-5:** For clarification: The wind shown in Figure 3 (Gaussian plume model) is not directly comparable to the wind shown in Figure 4 and 5 because the latter one is the geostrophic wind, whereas the former one is the wind at plume level(s),  correct?

**P19, Figure 6:** Which threshold was used for Figure 6, 500 ppm-m? Consider adding this information to the caption.

**P20, Figure 9:** Please add meaning of vertical bars to caption.

**P21, Figure 10:** Please add meaning of shaded area also to caption.

**P21, Figure 11:** Please add meaning of shaded area also to caption.

**P22, Figure 13:** Consider using a density plot for better visualization of the data cloud.

**Technical corrections**

**P1, L18:** "… Large Eddy Simulation …" → "… Large Eddy Simulations"

**P1, L29:** "… large geographical area …" → "… large geographical areas …"

**P2, L13:** "… at a resolution of 3-m …" → "… at a resolution of 3x3m² …" or "… at a resolution of 3 m …"

**P2, L20:** "… the retrievals measure the fine …" → "… the instrument observes the fine …"

**P3, L26:** "… approximately 15 minutes revisit time." → "… approximately 10 minutes revisit time." (compare to Figure 1)

**P5, L27:** "… the plumes structure." → "… the plume's structure."

**P6, L3:** LES is already defined in (P3, L5)

**P10, L11:** "… 1-SD error bars in the plot." → "… 1-SD error bars are shown in the plot."

**P10, L22:** "… large enough represent …" → "… large enough to represent …"

**References**

Cambaliza, M. O. L., Shepson, P. B., Bogner, J., Caulton, D. R., Stirm, B., Sweeney, C., Montzka, S. A., Gurney, K. R., Spokas, K., Salmon, O. E., Lavoie, T. N., Hendricks, A., Mays, K., Turnbull, J., Miller, B. R., Lauvaux, T., Davis, K., Karion, A., Moser, B., Miller, C., Obermeyer, C., Whetstone, J., Prasad, K., Miles, N., and Richardson, S.: Quantification and source apportionment of the methane emission flux from the city of Indianapolis, Elementa: Science of the Anthropocene, 3, 000 037, https://doi.org/10.12952/journal.elementa.000037, 2015.

Conley, S., Franco, G., Faloona, I., Blake, D. R., Peischl, J., and Ryerson, T. B.: Methane emissions from the 2015 Aliso Canyon blowout in Los Angeles, CA, Science, 351, 1317–1320, https://doi.org/10.1126/science.aaf2348, http://dx.doi.org/10.1126/science.aaf2348, 2016.

Frankenberg, C., Thorpe, A. K., Thompson, D. R., Hulley, G., Kort, E. A., Vance, N., Borchardt, J., Krings, T., Gerilowski, K., Sweeney, C., Conley, S., Bue, B. D., Aubrey, A. D., Hook, S., and Green, R. O.: Airborne methane remote measurements reveal heavy-tail flux distribution in Four Corners region, Proceedings of the National Academy of Sciences, 113, 9734–9739, https://doi.org/10.1073/pnas.1605617113, 2016.

Gordon, M., Li, S.-M., Staebler, R., Darlington, A., Hayden, K., O'Brien, J., and Wolde, M.: Determining air pollutant emission rates based on mass balance using airborne measurement data over the Alberta oil sands operations, Atmospheric Measurement Techniques, 8, 3745–3765, https://doi.org/10.5194/amt-8-3745-2015, 2015.

Hulley, G. C., Duren, R. M., Hopkins, F. M., Hook, S. J., Vance, N., Guillevic, P., Johnson, W. R., Eng, B. T., Mihaly, J. M., Jovanovic, V. M., Chazanoff, S. L., Staniszewski, Z. K., Kuai, L., Worden, J., Frankenberg, C., Rivera, G., Aubrey, A. D., Miller, C. E., Malakar, N. K., Tomás, J. M. S., and Holmes, K. T.: High spatial resolution imaging of methane and other trace gases with the airborne Hyperspectral Thermal Emission Spectrometer (HyTES), Atmospheric Measurement Techniques, 9, 2393–2408, https://doi.org/10.5194/amt-9-2393-2016, 2016.

Lavoie, T. N., Shepson, P. B., Cambaliza, M. O. L., Stirm, B. H., Karion, A., Sweeney, C., Yacovitch, T. I., Herndon, S. C., Lan, X., and Lyon, D.: Aircraft-Based Measurements of Point Source Methane Emissions in the Barnett Shale Basin, Environmental Science & Technology, 49, 7904–7913, https://doi.org/10.1021/acs.est.5b00410, 2015.

Tratt, D. M., Buckland, K. N., Hall, J. L., Johnson, P. D., Keim, E. R., Leifer, I., Westberg, K., and Young, S. J.: Airborne visualization and quantification of discrete methane sources in the environment, Remote Sensing of Environment, 154, 74 – 88, https://doi.org/10.1016/j.rse.2014.08.011, 2014.

---

## Referee Comment (RC2) · Anonymous Referee #2 · 19 Jun 2019

**General comments**

The manuscript "Towards accurate methane point-source quantification from high-resolution 2D plume imagery" by S. Jongaramrungruang et al. introduces a procedure to quantify the methane flux of a point source from a high resolution 2D imagery of the plume. Large Eddy Simulations are used to deduct the method. The flux inversion is described in detail and an error estimate for the method is given. The procedure is then applied to one case of a controlled release experiment, where it could reproduce the flux rate within the assumed error estimate.

The method seems useful, especially, as it does not need the wind speed as an additional input variable. All required values are only extracted from the 2D scene of the (vertically integrated) plume. This makes this method useful for optical measurements, and also for future satellite missions aiming at high resolution methane retrievals.

The method is novel and clearly outlined, the paper fits well into the scope of AMT. The manuscript is well structured, however, sometimes (long) sentence structures made it hard for me to follow. It would be nice, if the authors invest some time for rephrasing, giving the reader a more fluent reading experience.

The authors should use SI units in the preferred inverse notation throughout the manuscript (as stated in AMT manuscript guidelines), several times ppm-m is used (in the text and figure labels), which may be ppm m in SI units (?).

For better understanding, the authors should avoid synonyms (e.g. synthetic measurements vs. pseudo measurement vs. synthetic observation).

Generally, many of the figure legends, axis labels, and other labels might be to small for a good reproduction in the final publication.

Please, do not forget to add the necessary sections: "Data and code availability", "Author contributions", "Conflict of interest".

I recommend publication in AMT, subject to some improvements.

**Specific comments**

**p. 3, ll. 6ff.:** The inclusion of retrieval noise in the simulation of synthetic measure-ments is mentioned, but I have not found anything about this topic in Sec. 4.1 or Sec. 5, which describe more details. Is additional noise used (and if, which) in preparing the synthetic measurements? Is this affecting the estimated errors (and how)? Maybe more details can be included in Section 4.2.

**p. 3, l. 26:** A 15 minute revisit time is mentioned, sub-figures of Fig. 1 show in-strument overpasses in time intervals of 7 to 9 minutes (as also stated in the figure caption).

**p. 6:** I recommend to restructure Section 4, as Section 4 itself is empty. It could rather be: "4 Large Eddy Simulation" and "5 Synthetic measurement", following Sec-tions change accordingly. The section on synthetic measurement could be extended by some details on the additional noise added. Maybe the applied detection thresholds could also be included here, instead of in Section 2.

**p. 6, l. 14:** In this study latent and sensible surface heat fluxes are kept con-stant.
Was the method tested with other settings (except of the controlled release exper-iment)? What would happen if these fluxes are varied? How would the plume be affected? Would this impact the method? If the method is still applicable with the derived correlations, would the error estimates change?
Some of these questions are answered in "Discussion and conclusion" and answers may not fit in "LES setup". The authors should consider a reference to the discussion section and a strengthening of the corresponding paragraph there, or an additional section on limitations.

**p. 8, ll. 4ff.:** If I understand it right, the instantaneous value for U-10 is written out from the LES simulation every minute. When the plume structure is – as stated

in line 7 – influenced by the wind during this minute, the LES model (integration) time-step should be smaller. The model time-step should be somewhere around few seconds, taking into account the high resolution of 5 m. However, I found no value for the model time-step, maybe you could include it in the Section about the LES setup. Maybe a rephrasing of the sentence in ll. 4f. could help to understand that instantaneous wind values are written out from the model simulation. Also, using the term "timestep" for one instance of the model output may a bit misleading, as output does not coincide with every model time-step, maybe you could use "snapshot".

**p. 9, ll. 17ff.:** For the method to work, has the plume origin to be known for the angular mass binning? For LES simulation and field campaigns this should be no problem (when measuring unknown flux rates from a known point source). Also, for flux inversions of known sources from future high resolution satellite imagery, e.g, for emission monitoring, the method will be useful. In other situations, it might be more challenging. Limitations of the method should be addressed in the discussion section, or an additional section on limitations. (e.g., more than one point source or parts of a second plume included in a scene ...)

**p. 10, ll. 16f.:** The paragraph about error estimates could be strengthened. Please do not just give the value of $\chi^2$ without interpreting this result. Maybe additional description how the "average percentage error" is calculated, and how the error propagates through all steps of the method could be added.

**p. 10, ll. 26ff.:** Additional information about the controlled release experiment would be nice. Maybe, include a figure of the scene, or a figure of the angular distribution (comparable to Fig. 8)...

**p. 12, ll. 3ff.:** This paragraph addresses the limitations arising from using constant sensible and latent surface heat fluxes.

The authors should consider to present the limitations in a separate section. The last sentence of the paragraph should be moved closer to the discussion of the constant surface fluxes. All limitations should be named and discussed.

**p. 20, Fig. 8:** The figure contains two black lines, which are not explained, I assume they denote something like the main axis of the plume.
If the plume is already rotated, why is the maximum of the normalized angular IME distribution not at 0?

**Technical corrections**
**p. 1, l. 18:** "m/s" → "m s$^{-1}$"
**p. 1, l. 19:** "kg/hr" → "kg h$^{-1}$"
**p. 1, l. 20:** "5m" → "5 m"
**p. 2, l. 2:** "2$^{nd}$" → "second"
**p. 2, l. 2:** "... in the Earth's atmosphere, ..." → "... in Earth's atmosphere, ..."
**p. 2, l. 6:** "facilities scale" → "facility scale"
**p. 2, l. 7:** "... while the in-situ measurements ..." → "... while in situ measurements ..."
**p. 2, l. 8:** "Improved estimates of the CH$_4$ emissions at this point-source scale is critical ..." → "Improved estimates of the CH$_4$ emissions at point-source scale are critical ..."
**p. 2, l. 13:** "3-m" → "3 m"
**p. 2, l. 14:** "... in the direct vicinity ..." → "... in direct vicinity ..."
**p. 2, l. 16:** "kgCH$_4$/hr" → "kg(CH$_4$) h$^{-1}$"
**p. 2, l. 18:** "molecule/cm$^2$" → "?"
**p. 2, l. 29:** "... inferred from 10-m wind speed by in-situ ..." → "... inferred from 10 m wind speed by in situ ..."
**p. 3, ll. 6ff.:** "Using 3D LES model output for each time-step, we simulated synthetic 2D airborne measurements by applying the respective averaging kernels as well as retrieval noise (Section 5-5.1)." → "Using 3D LES model output of each model time-step, we simulated synthetic 2D airborne measurements by applying the respective

averaging kernels as well as retrieval noise (Sec. 4.2)."

**p. 3, l. 9:** "... imply the wind speed from the plume spatial distribution ..." → "... deduce the wind speed from the plume's spatial distribution ..."

**p. 3, l. 22:** "... & ..." → "... and ..."

**p. 3, l. 26:** "... 15-minute revisit time ..." → "... 15 minute revisit time ..."

**p. 3, ll. 26f.:** "Evidently, the plumes are changing in time and exhibit fine-scaled features due to atmospheric turbulence." → "Evidently, the plume is changing in time and exhibits fine-scaled features due to atmospheric turbulence."

**p. 4, l. 1:** "Figure 1" → "Fig. 1"

**p. 4, l. 2:** "5 *n*m" → "5 nm"

**p. 4, l. 3:** "... (Averaging Kernel) of one at ..." → "... (averaging kernel) of 1 at ..."

**p. 4, l. 3:** "Figure 2" → "Fig. 2"

**p. 4, ll. 5ff.:** "It has varying sensitivities in the vertical layer, each of which can be calculated as the derivative of the retrieved total column amount with respect to the change in that particular layer." → "Its varying sensitivity in the vertical can be calculated as the derivative of the retrieved total column amount with respect to the change in a particular layer." ?

**p. 4, ll. 10:** "... horizontal grid (i,j), $\Delta x$, $\Delta y$ ..." → "... horizontal grid cell (i,j). $\Delta x$, $\Delta y$ ..."

**p. 4, ll. 17f.:** "On the other hands, ..." → "On the other hand, ..."

**p. 4, ll. 18f.:** "... even larger than one ..." → "... even larger than 1 ..."

**p. 4, ll. 19f.:** Can you please rephrase the second part of the sentence: "This means that the instrument is almost blind to methane near the ground, while the enhancement at higher levels can be amplified to be even more than the actual methane amount in the column."?

**p. 4, ll. 25f.:** "Here, we use an average constant threshold value at 500 ppm-m (or 1.34 $10^{18}$ molecules/cm$^2$), which is a common range for AVIRIS-NG." → "Here, we use a constant threshold of 500 ppm m (or 1.34 $10^{18}$ cm$^{-2}$), which is a common value for AVIRIS-NG."

**p. 4, l. 27:** "Gaussian Plumes Modelling & Its Limitations" → "Gaussian plume modelling and its limitations"

**p. 4, l. 30, p. 5, l. 1:** "The dispersion functions depend on the atmospheric stability classification (e.g., Pasquill, 1961)." → "The dispersion functions depend on the atmospheric stability."

**p. 5, l. 3:** "The three-dimensional Gaussian plume equation is given by (Eq.2, Matheou and Bowman, 2016)" → "The three-dimensional Gaussian plume equation is given by (Matheou and Bowman, 2016)"

**p. 5, ll. 8f.:** "By integrating Equation 2 in the Z-direction, the methane column enhancement can be modelled in analytical form as (Eq.3)" → "By integrating Eq. 2 in z-direction, the methane column enhancement can be modelled in analytical form as"

**p. 5, l. 11:** "Based on this model, we can vary source rates, wind speeds, and stability categories to simulate the 2D concentration field." → "Based on this model, we can vary source rate, wind speed, and stability category to simulate the 2D integrated concentration field."

**p. 5, l. 13:** "300 kg/hr" → "300 kg h$^{-1}$"

**p. 5, l. 27:** "... plumes structure." → "... plume structure."

**p. 6, l. 1:** "Large-Eddy Simulation and Pseudo-Measurement" → "Large Eddy Simulation and synthetic measurement"

**p. 6, l. 2:** "LES Set Up" → "LES setup"

**p. 6, l. 4:** "... 3-dimensional spatial CH$_4$ distribution ..." → "... 3-dimensional CH$_4$ distribution ..."

**p. 6, l. 5:** "... currently available from aircraft ..." → "... currently available from aircraft measurements ..."

**p. 6, l. 10:** "... mixed layer-inversion-free troposphere structure with ..." → "... mixed layer inversion free troposphere with ..."

**p. 6, l. 12:** "The inversion is $\Delta\theta/\Delta z = \frac{12}{100}\,K\,m^{-1}$" → "The lapse rate is $\Delta\theta/\Delta z = 0.12\,K\,m^{-1}$"

**p. 6, l. 13:** "m/s" → "m s$^{-1}$"

**p. 6, l. 14:** "... heat fluxes are 400 and 40 W/m$^2$, based on the typical field campaign

data." → "... heat fluxes are 400 and 40 W m$^{-2}$, based on typical field campaign data."
**p. 6, l. 16:** "The model domain is 10.24 x 2.56 x 1.5 km in the x, y, and z directions and ..." → "The model domain is 10.24 x 2.56 x 1.5 km$^3$ in the x, y, and z direction and ..."
**p. 6, l. 19:** "Furthermore, the 10-m and 2-m wind speeds ..." → "Furthermore, the 10 m and 2 m wind speeds ..."
**p. 6, l. 23:** "... LES runs ..." → "... LES simulations ..."
**p. 6, l. 27:** "This allow us ..." → "This allows us ..."
**p. 7, ll. 3ff.:** "The left column of Figure 4 represents a single snapshot of plumes, while the right column shows the time-averaged plumes from an ensemble of 60 timesteps, spanning a duration of 60 sequential minutes in total, under distinct background wind speeds but with the constant flux rate." → "The left column of Fig. 4 shows single snapshots of the plume, while the right column shows the time-averaged plume snapshots over 60 timesteps, spanning a duration of 60 sequential minutes in total, under distinct background wind speeds but with a constant flux rate."
**p. 7, ll. 6f.:** "The ensemble averages in the right column ..." → "The temporal averages in the right column ..."
**p. 7, l. 13:** "... evident in the plume snapshots as well as their ensemble means as shown in Figure 4." → "... evident in the plume snapshots as well as in their ensemble means shown in Fig. 4."
**p. 7, ll. 14f.:** "... synthetic observations from AVIRIS-NG and those from HyTES over the same plume ..." → "... synthetic measurements for AVIRIS-NG and HyTES of the same plume ..."
**p. 7, l. 16:** "... the measurements from HyTES ..." → "... the synthetic measurements for HyTES ..."
**p. 7, l. 18:** "... resulting in the plume ..." → "... resulting in a plume ..."
**p. 8, l. 2:** "... 2-m (U-2) or 10-m (U-10) ..." → "... 2 m (U-2) or 10 m (U-10) ..."
**p. 8, ll. 10f.:** "... as the ideal case of having continuous U-10 output, it has been found in our work to provide a robust correlation with the overall pattern of the plume (see Section 5.2)." → "... as the ideal case of having continuous U-10 output, it provides a

robust correlation with the overall pattern of the plume (see Section 5.2)."

**p. 8, ll. 17f.:** "For each wind speed and flux rate, we have 60 timesteps of methane plumes from the LES model output, each with one minute apart." → "For each wind speed and flux rate, we have 60 snapshots of methane plumes from the LES model output, with an temporal interval of one minute."

**p. 8, l. 18:** "... across these ensembles." → "... across these snapshots."

**p. 8, l. 19:** "Although the shape of the plumes can vary ..." → "Although the shape of a plume can vary ..."

**p. 8, ll. 23f.:** "The mean values corresponding to various background wind speed and flux rate are plotted in Figure 6." → "The mean values corresponding to various background wind speeds and flux rates are plotted in Fig. 6."

**p. 8, ll. 28f.:** "The non-linearity can be explained from the fact that we impose the detection threshold value to mask out the plume." → "The non-linearity can be explained from the fact that we impose a detection threshold to mask out the plume."

**p. 9, l. 1:** "kg/hr" → "kg h$^{-1}$"

**p. 9, ll. 5f.:** "This is a primary reason why the IME varies with the flux rate with different degree of non-linearity at different wind speeds as found in Figure 6." → "This is the primary reason why the IME varies with the flux rate with different degree of non-linearity at different wind speeds as found in Fig. 6."

**p. 9, ll. 6ff.:** "The background wind speed is the integral component that drives the spatial distribution of the plume and entangles the IME and the flux rate inversion." → "The background wind speed is the integral component that drives the spatial distribution of the plume and correlates the IME with the flux rate."

**p. 9, l. 11:** "... reanalysis weather data." → "... reanalysis data."

**p. 9, l. 24:** "... 10-1000 kg/hr." → "... 10-1000 kg h$^{-1}$."

**p. 9, l. 26:** "... of the plume at high wind speed case, i.e. 10 m/s, is narrower..." → "... of the plume at highest wind speed of 10 m s$^{-1}$ is narrower ..."

**p. 9, l. 31:** "... time-snapshots ..." → "... time snapshots ... / ... temporal snapshots ..."

**p. 10, l. 2:** "... with the IME in the plume as explained earlier in Figure 6, ..." → "... with

the variation of IME with flux rate (Fig. 6), ..."

**p. 10, l. 6:** "... , we can draw a line to estimate ..." → "... , we can estimate ..."

**p. 10, l. 11:** "... wind speed, both the mean value and the uncertainties from the lower and upper 1-SD error bars in the plot." → "... wind speed, the mean value and the uncertainties from the lower and upper estimate of 1 standard deviation."

**p. 10, l. 18:** "... this method permits estimation of emission magnitude." → "... this method permits estimation of total emission flux rate."

**p. 10, l. 22:** "The size of 30 is chosen arbitrarily but is large enough represent a situation when we consider the total fluxes from a region." → "The sample size of 30 is chosen arbitrarily but is large enough to represent a situation for the estimation of total fluxes from a region." ?

**p. 10, l. 23:** "The comparison between the predicted and the actual sum of 30 plumes aggregation is shown in Figure 13." → "The comparison between the predicted and the actual total flux aggregated over 30 plumes is shown in Fig. 13."

**p. 10, l. 24:** Please rephrase: "Most of the aggregation predictions lie on 1-to-1 line, implying that there are no significant systematic biases in our method."

**p. 10, l. 25:** "The mean percentage error of all these aggregates is 2.9**p. 10, ll. 26f.:** "To further demonstrate the validity of this method, we applied this to a controlled release experiment from a natural gas pipeline with the flux rate of $89 \pm 4$ kg/hr." → "To further demonstrate the validity of this method, we applied it to a controlled release experiment from a natural gas pipeline with a flux rate of $89 \pm 4$ kg h$^{-1}$."

**p. 10, ll. 27ff.:** "Based on a sample from the actual AVIRIS-NG scene over the source location, we measured the IME, constructed the angular distribution of the plume to obtain the width to predict the wind speed and hence predicting the flux rate." → "Based on a sample of the actual AVIRIS-NG scene over the source location, we calculated the IME, constructed the angular distribution of the plume to obtain its width to deduce the wind speed and hence predicting the flux rate."

**p. 10, ll. 29f.:** "The value that we predict is $118 \pm 30$ kg/hr, consistent with the actual release flux within an error estimate." → "The value that we predict is $118 \pm 30$ kg h$^{-1}$,

consistent with the actual release flux within the error estimate."

**p. 11, l. 24:** "Given that point source are..." → "Given that point sources are..."

**p. 11, l. 25:** "... single-measurement ..." → "... single measurement ..."

**p. 11, l. 30:** "... validated from the application ..." → "... validated by the application ..."

**p. 11, l. 32:** "This provides great values in ..." → "This provides added value in ..."

**p. 12, l. 1:** "... point sources emissions ..." → "... point source emissions ..."

**p. 12, l. 3:** "... LES runs ..." → "... LES simulations ..."

**p. 12, ll. 8f.:** "For our purpose, we set the threshold value to 500 ppm-m throughout our study to match the current capability of the current instrumentations." → "In this study, we set the threshold to 500 ppm m to match the capabilities of the current instrumentations."

**p. 12, l. 11:** "Repeat overflights that results in multiple snapshots of the same source can also further reduce the uncertainties from the transient variations of the plume due to turbulence." → "Repeat overflights that result in multiple snapshots of the same source can also further reduce uncertainties from transient variations of the plume due to turbulence."

**p. 12, ll. 14f.:** "These methods could be applied ..." → "This method could be applied ..."

**p. 15, Fig. 2:** "A plot showing column averaging kernels ..." → "Column averaging kernels ..."

**p. 16, Fig. 3:** Labels are too small.
"ppm-m" → "ppm m"
"kg/hr" → "kg h$^{-1}$"

**p. 17, Fig. 4:** Labels are too small.
"ppm-m" → "ppm m"
"m/s" → "m s$^{-1}$"
"kg/hr" → "kg h$^{-1}$"
"... detection threshold is set at 500 ppm-m." → "... detection threshold set to 500 ppm m."

"Note that the average did not reach a true ensemble average yet as sample size was finite (i.e. the average still exhibit fine structure)." → "Note that the temporal averages do not reach a true ensemble average as sample sizes are finite (i.e. the averages still exhibit fine structure)."
**p. 18, Fig. 5:** Labels are too small.
"ppm-m" → "ppm m"
"kg/hr" → "kg h$^{-1}$"
"... when observed by AVIRIS-NG instrument ..." → "... when applying the AVIRIS-NG instrument ..."
"... but instead observed with HyTES." → "... but applying the HyTES averaging kernel."
"... detection threshold is set at 500 ppm-m." → "... detection threshold is set to 500 ppm m"
**p. 19, Fig. 6:** "kg/hr" → "kg h$^{-1}$"
"m/s" → "m s$^{-1}$"
**p. 19, Fig. 7:** "kg hr−1" → "kg h$^{-1}$"
"ppm-m" → "ppm m"
**p. 20, Fig. 8:** Labels are too small.
"ppm-m" → "ppm m"
"m/s" → "m s$^{-1}$"
"kg/hr" → "kg h$^{-1}$"
"... its angular distribution across the plume (right) and its Cartesian distribution along the plume (top)." → "... its angular distribution of IME across the plume (right) and its Cartesian distribution of IME along the plume (top)."
**p. 20, Fig. 9:** Please, include description of error bars in caption.
**p. 21, Fig. 10:** Please, include description of shading.
"... averaged over time steps and flux rates." → "... averaged over snapshots and flux rates." ?
**p. 21, Fig. 11:** Please, include description of shading.

"... flux rates and wind speeds ..." → "... flux rate and wind speed ..."
**p. 22, Fig. 12:** "kg hr$^{-1}$" → "kg h$^{-1}$"
"methods" → "method"
Please, include description of error bars.
**p. 22, Fig. 13:** "kg hr$^{-1}$" → "kg h$^{-1}$"

––––––––––––––––––––––––––––––

---

## Author Comment (AC3) · 31 Aug 2019

We have broken down some sentences to be shorter for the sake of readability, adjusted some labels in the figure for more readability and have added the "Data and code availability","Author contributions", "Conflict of interest" sections to the manuscript.

---

## Author Response (AR1)

**Response to Referee 1**

**General comments**

The publication "Towards accurate methane point-source quantification from high-resolution 2D plume imagery" by Jongaramrungrang et al. deals with the important aspect of inverting atmospheric concentration gradients to fluxes or emission rates of not well-constraint CH4 point sources. In order to invert these observations, wind information in the measured area is needed and equally important as the concentration measurements themselves. However, as outlined in the introduction of the manuscript, acquiring wind information, preferably, simultaneously and at an adequate spatial and temporal resolution can be a challenging task. Therefore, the authors propose a new method, which solely relies on spatially resolved imaging data from airborne remote sensing instruments and high resolution large eddy simulations (LES)to estimate the prevailing wind speed during the time of the overflight.The wind speed is then derived from the shape of the observed plume and used during the inversion process to compute a flux of the investigated sources.The described method is a novel and promising approach to quantify CH4 point source emissions from aircraft but also potentially from space without having to rely on real wind measurements. The manuscript fits well in the scope of AMT and I recommend publication after some modifications along the line of the comments below. In general, the manuscript is well written. The method is described in a comprehensible way, however, some additional information concerning the figures would improve their readability (see also specific comments). Additionally, the authors should add some more information regarding the stated errors and their propagation to the predicted fluxes(see also specific comments).In my opinion, the manuscript could be strengthened by adding more extensive comparisons and applications to real data. So far, most parts of the approach were developed based on (theoretical) model (LES) studies, whereas only few real observations were used to support the novel approach. Therefore, I would recommend to expand on the already given examples in the manuscript:(1) controlled release experiment and(2) analyses of overflights shown in Figure 1 (see also specific comments).

We thank the reviewer for the constructive comments and appreciate the thoughtful review.

**Specific comments**

P2, L12:Consider to also add publications regarding the HyTES instrument(e.g., Hulley et al., 2016) or other imaging instruments, which also have CH4 point sources successfully detected, e.g., MAKO (Tratt et al., 2014).

We added Hulley et al. 2016 and Tratt et al. 2014 accordingly to cite publications for HyTES and MAKO instruments.

P2, L24f:I would suggest to either only focus on remote sensing studies (by satellite and aircraft)and remove Conley et al., 2016,or to better distinguish between remote sensing and in-situ studies. If in-situ studies are included I would also recommend to add, e.g., Cambaliza et al., 2015,Gordon et al., 2015, and Lavoieet al., 2015.,which have also performed extensive analyses regarding airborne in-situ observations and resulting fluxes.

We would like to mention other studies done by an airborne in-situ approach using a mass balance calculation based on the enhancement downwind of the source. We added the citations from Conley et al., 2016; Jacob et al., 2016,  Cambaliza et al., 2015,Gordon et al., 2015, and Lavoieet al., 2015.

P3, L6:"... under various background wind speeds and surface heat fluxes.": Later in the manuscript (P6, L14), it is stated that only one value for the heat fluxes was used("The surface sensible and latent heat fluxes are 400 and 40 W/m².").

We have conducted sensitivity analysis for other values of latent and surface heat fluxes and added an additional small section on this to show that the changes in sensible and latent heat fluxes do not significantly impact the results and are less crucial than wind speed. We added an additional section (6.4) into the text.

P3, L11:"... and presence of ancillary information on the actual wind speed(Section 5.3).": Based on this statement, I would expect to see a comparison between the derived wind speed based on the new method and actual wind observations or reanalysis data in Section 5.3., however, I was not able to find a paragraph referring to actual wind speed observations or reanalysis data.

We added the result of the predicted wind compared to the value from controlled release experiment to section 6.3 P11 L5. The geostrophic wind speed is predicted to be 3.3 +- 1.2 m/s, compared to the surface sonic wind at the source measured at 1.9 m/s. This is consistent given that geostrophic wind is typically higher than the surface wind speed.

P4, L1:Why not also showing a real example observation of HyTES. Maybe, there is one available for the same scene as shown in Figure 1. If this is the wrong place, because in Figure 1, the authors intend to show the high variability of the plume structure during multiple overflights, I would recommend, if available, to add a comparison of an AVIRIS-NG and HyTES observation after Figure 5, where simulated plumes observed by the two instruments are compared.

We added a scene from HyTES at the same location within approximately the same time for this comparison. However, this will be restricted to Clutter Matched Filter retrievals, which are similar but not identical to the Optimal Estimation based retrieval for which the averaging kernels were computed.

P4, L1ff:This would also be a good place to add some more information regarding the applied retrieval algorithms.As Frankenberg et al., 2016, are already cited multiple times and the plume shown in Figure 1 (bottom right) appears to be similar to the one inFigure S1 in Frankenberg et al., 2016, I assume that either the matched filter technique or the IMAP-DOAS method, described in that publication, has been used to retrieve the columns shown in Figure 1 or the ones from the controlled release experiment at the end of Section 5.3used for a flux inversion.Similar for the HyTES algorithm if real observations are added.

For AVIRIS-NG scenes illustrated in Figure 1 the linearized matched filter technique was used. For HyTES, Clutter Matched Filter Approach (CMF) was used, as explained in the earlier comment.

P4, L26:What is the basis of the chosen threshold of 500 ppm-m? As stated in (P3, L30), it is connected to the measurement precision of the AVIRIS-NG instrument and is thus a property of the instrument (Why does it then differ from the value of 200 ppm-m applied in Frankenberg et al., 2016, Section 'IME' in 'Materials and Methods'?). Furthermore, what is the reasoning in using the same threshold also for HyTES simulations, e.g., as shown in Figure 5?

The 500 ppm-m has been chosen as a somewhat conservative threshold, close to what we expect as single-measurement noise for typical scenes. In Frankenberg et al, a lower threshold was used but data were also smoothed and plume segmentation algorithms applied. As for HyTES, we used the same threshold just to exemplify the differences due to averaging kernels only, as opposed to thresholds. We will make this clear in the revised version. For now, we don't have a unique quantitative detection threshold for HyTES, which would strongly depend on surface temperature and other variables. Thus, it is mostly for illustration at the moment.

P5, L5:Please add some explanation for the 'sum' and the parameter 'm' in Equation 2 to the text.
In the Gaussian plume model, we assume equilibrium to find the concentration of plume in 3-dimensional space. For the vertical direction, when we assume an inversion height at $Z_i$, the model can assume reflective boundary condition. The parameter m multiplied by $Z_i$ indicates the height that the reflection occurs and the summation over this parameter m shows the total concentration at each height within 0 to $Z_i$. We added this explanation to the text.

P6, L10-15:What is a reasonable range of the proposed values (initial inversion height, potential temperature,specific humidity, surface sensible and latent heat flux) for initializing the LES model? Are these educated guesses or are they based on actual field or reanalysis data? Given these ranges, the authors could also verify their assumption "... our method ... should not be significantly impacted ..." in the conclusions (P12, L6) by performing various LES runs using different initial values. I would agree with the authors that the column-integrated enhancements are not significantly influenced by the surface heat fluxes if the threshold were 0 ppm-m. However, I am curious if and how Figure 6 and 7 might change under various initial conditions, or whether they are well within the error bars.

These values are based on typical field campaign data. To show how the result from our method applies to the field of different conditions, we have added additional LES runs with different combination of sensible and latent heat fluxes (SH and LH) in two more cases: SH = LH, LH > SH to compared with the typical condition we use in this paper (SH > LH)

[Figure]

We found that the relationship between observed IME, flux rate and wind speed under new conditions (orange and blue lines) lie within 1-SD error from our original condition (green line). This shows that the uncertainties associated with the change in these conditions will not significantly impact our method and are captured well with the range of errors we have analysed.

Based on these results, we added a small section on this discussion.

P8, L27:Do the authors only mean Frankenberg et al., 2016, by "... has been ignored in previous studies." or are there further ones?

Previous studies in this case refer to both Frankenberg et al., 2016 and Varon et al,. 2018 both of which assume flux is linearly proportional to IME as mentioned in the introduction.

P9, L33:Could the authors add the fitted polynomial also to Figure 10? Why do the authors use a fifth-degree polynomial? Is there a physical relationship, which relates wind speed and plume angular width by a fifth-degree polynomial or is it just the 'best' fit? Could the relationship in Figure 10 also be explained by an exponential curve? Assumption: The relation in Figure 10 is determined by averaging over an ensemble of LES realizations. If this is done, I expect the resulting plume to be approximately Gaussian as seen in Figure 4, e-f.
We added the fitted curve to Figure 10 and its caption. We performed additional experiments and found that exponential fit provide a good fit to the curve without overfitting.

P10, L5-13:Basically, in this paragraph the authors summarize their developed method. I have some questions/comments to the used example.a) The error bars (shaded area) shown in Figure 11:Are they related to the errors shown in Figure 6 or to Figure 10? b) Have the flux rates, shown in Figure11, been corrected for the missing IME (as indicated in Figure 7)? c) How is the error (1-SD of the plume angular width) shown in Figure 10 related/translated to the wind speed error, which then linearly propagates to the estimated flux?

 a) The error bars from Figure 11 are directly related to Figure 6. For a given IME, we found the possible range of fluxes for a given value of wind speed from the relationship from Figure 6.
b) Yes, the flux rate in Figure 11 is based on the relationship from Figure 6, in which the IME is the IME that is observed on the scene after applying the threshold.
c) Based on Figure 10, at a given angular width measured from the plume, we can predict the wind speed from the fitted line. The associated uncertainties of the wind speed are approximated by the possible range within 1-SD area. We assume that by projecting a value of plume width onto the corresponding range of wind within 1-SD shading area, we obtain uncertainties for predicted wind speed that approximately represent 1-SD error for the wind speed as well.

We modified the text in this paragraph accordingly in the manuscript.

P10, L14-17: Could the authors be more precise in terms of the given "average percentage error" of 30%? I assume it is the mean value of the vertical error bars shown in Figure 12.However, as in reality not only entire fields/regions of CH4 sources (as then investigated in P10, L18-25)are investigated but also single plumes, which are typically observed only once, an interesting measure would also be the average of the absolute differences (also in percent) of the predicted flux rates and the corresponding prescribed flux rates.This would give an idea of the magnitude of the bias one can expect from the method.I assume that the observed bias in the flux for the controlled release experiment of ~32% lies within this computed theoretical value.

The average percentage error here is meant to be the average of the percentage differences between the predicted value and the actual value in Figure 12. Each point in Figure 12 is showing prediction for a single plume measurement, not for the entire field.
To make this clearer, we have accordingly revised the sentence to be "The average of the percentage differences (in absolute terms) between the predicted value and the actual value for single point source predictions is approximately 30%,..."

P10, L25: What do the authors exactly mean by "mean percentage of error" ? Is that the average of all differences between actual and predicted flux OR is it a similar error as computed for Figure 12 in P10, L14-17? If the authors refer to the latter one, I would suggest to also compute the average of all differences between actual and predicted flux (not the average of the absolute differences as in the previous comment). For example, a positive or negative value would then quantify an over-or underestimation caused by the method on average. The same exercise can be done for Figure 12 because it appears (as for Figure 13) that more predicted fluxes lie above the 1-to-1 line then below especially for larger fluxes.

We computed both the mean of absolute differences from all these aggregates which is 5.1% with a standard deviation of 3.9%, and the average of all differences (negative and positive) which results in 2.9% with the standard deviation of 5.9%. We added this result to this section.

P10, L26ff: The possibility to compare the novel approach, which is mostly based on 'theoretical' models,to real data is a huge strength of the publication. Therefore,I would recommend to expand this part of the publication. Some starting points are already given.First of all, the authors could add some more information regarding the already analyzed controlled release experiment allowing for a better judgement by the reader.Useful information would be (a)a figure showing the overflight and the retrieved plume and CH4 column enhancements(similar to Figure 1), (b)the fitted wind speed, which is then used to invert the IME to a flux, and (c)as the observation is based on a controlled release experiment, do the authors have access to real wind observations on-site or at least to meteorological reanalysis data, which can then be compared to the fitted wind to test its plausibility? Additionally, the authors nicely show multiple overflights of one source within ~25 minutes by the AVIRIS-NG instrument in Figure 1. It would be an interesting opportunity to apply the developed method to the four overflights shown in that figure and discuss the resulting fitted winds and inverted fluxes.

We added more information regarding the controlled release experiment. The information included are
(1) a figure (Fig.15) showing the actual overflight AVIRIS-NG scene
(2) the deduced wind speed that was then applied to calculate the flux rate

(3) comparison between the deduced background wind speed and the available observation at 10-m during the controlled release experiment flight.
We also applied our method to the observed AVIRIS-NG scenes from Fig.1.

P10, L29:I assume the given estimated emission of 118 kg/hr is already corrected by the potentially missed IME as indicated in Figure 7, right? Additionally, could the authors elaborate on what error sources are included in the error estimate (of 30kg/hr) of the predict flux.
Yes, it has been corrected. In our method, we apply the observed IME to the relationship in Figure 6 which has the relationship between observed IME, wind speed and flux rates. The geostrophic wind speed is predicted to be 3.3 +- 1.2 m/s. The error from the predicted wind speed results on the error of 30 kg/hr. We have added this details in this section.

P15, Figure 2:Please clarify whether altitude on z-axis is given in meters above sea level or above ground level. Additionally, please harmonize the minimum altitudes of the computed averaging kernels, either to 0 m or to a specific surface elevation. Consider also adding the aircraft altitude(s) which the examples CAKs are valid / have been computed for.
The altitude on z-axis is given above ground level. We harmonized the minimum altitude on the plot. In the Thermal case (HyTES) the flight altitude is an important factor for the CAK. In Figure 2, the CAK of HyTES was computed for an altitude of about 3 km. For the shortwave range, however, the CAK of AVIRIS-NG is not impacted significantly by flight altitude. In this paper, we focus our analysis on the results from AVIRIS-NG scenario. We added this info to the caption of Fig.2.

P16, Figure 3:Consider adding the true IME (idealized threshold of 0 ppm-m) to the caption.Additionally, consider adding labels for the stability classes to the caption so that the reader sees immediately their meaning without looking up the relevant information in the cited publications, e.g.,A = very unstable; B = moderately unstable; …
The idealized IME when threshold is 0 ppm-m has been added accordingly. Without any threshold, the IME actually depends on the box size as enhancements approach the domain boundaries. In this case, we calculated the IME that would have been observed within the box as shown in the figure if the threshold were to be 0. We have added the meaning for each label as A=very unstable, B=unstable, C=slightly unstable. We have added this description to the caption of Figure 3 for more clarity.

P17, Figure 4:Consider adding the true IME (idealized threshold of 0 ppm-m) to the caption (as suggested forFigure 3),and the variance of IME(of the 60 individual snapshots)to each plot in the right column for the three cases of wind speed so that the reader can assess the statement from (P7, L10f).

Based on the values of IME across snapshots, we computed the variance of IME, and have added these values to the right column panels. We have also added the idealized IME similar to what was described above in "P16, Figure 3."

P18, Figure 5:Consider adding the true IME (idealized threshold of 0 ppm-m) to the caption (as suggested forFigure 3)
We have also added the idealized IME similar to what was described above in P16, Figure 3.

Figure 3-5:For clarification: The wind shown in Figure 3 (Gaussian plume model) is not directly comparable to the wind shown in Figure 4 and 5 because the latter one is the geostrophic wind, whereas the former one is the wind at plume level(s), correct?
Yes, that is correct. We added this clarification to the Figure captions.

P19, Figure 6: Which threshold was used for Figure 6, 500 ppm-m? Consider adding this information to the caption.
Yes, we have used the threshold of 500 ppm-m in the analysis of Figure 6. We have added this information to the caption of Figure 6.

P20, Figure 9:Please add meaning of vertical bars to caption.
The vertical bars represent the standard deviation of the normalized IME at a given angle across all snapshots.  We added this detail to the caption of Figure 9.

P21, Figure 10:Please add meaning of shaded area also to caption.
The shaded area represents one standard deviation from the mean plume angular width for each wind speed. The standard deviation is computed across different values of flux rates and snapshots. We have added the meaning of the shaded area to the caption of Figure 10.

P21, Figure 11:Please add meaning of shaded area also to caption.
The shaded area represents one standard deviation for the flux rate at a particular value of IME for each wind speed. The 1-SD for the flux rate is approximated by the possible range of flux rates resulted from an observed IME and wind speed in Figure 6. I added the meaning of the shaded area to the caption of Figure 11.

P22, Figure 13:Consider using a density plot for better visualization of the data cloud.
Because we would like to illustrate the comparison between predicted and actual values at each point, showing the data cloud could be more helpful in our case.

Technical corrections

P1, L18:"... Large Eddy Simulation..." "... Large Eddy Simulations"
P1, L29:"... large geographical area ..." "... large geographical areas ..."
P2, L13:"... at a resolution of 3-m ..." "... at a resolution of 3x3m² ..." or "... at a resolution of 3 m ..."
P2, L20:"... the retrievals measure the fine ..." "... the instrument observes the fine ..."
P3, L26:"... approximately 15 minutes revisit time." "... approximately 10 minutes revisit time." (compare to Figure 1)
P5, L27:"... the plumes structure." "... the plume's structure."
P6, L3:LES is already defined in(P3, L5)
P10, L11:"... 1-SD error bars in the plot." "... 1-SD error bars are shown in the plot."
P10, L22:"... large enough represent ..." "... large enough to represent ..."
We corrected all of these technical corrections accordingly.

**Response to Referee 2**

The manuscript "Towards accurate methane point-source quantification from high-resolution 2D plume imagery" by S. Jongaramrungruang et al. introduces a procedure to quantify the methane flux of a point source from a high resolution 2D imagery of the plume. Large Eddy Simulations are used to deduce the method. The flux inversion is described in detail and an error estimate for the method is given. The procedure is then applied to one case of a controlled release experiment, where it could reproduce the flux rate within the assumed error estimate.

The method seems useful, especially as it does not need the wind speed as an addi-tional input variable. All required values are only extracted from the 2D scene of the(vertically integrated) plume. This makes this method useful for optical measurements,and also for future satellite missions aiming at high resolution methane retrievals.

The method is novel and clearly outlined, the paper fits well into the scope of AMT. The manuscript is well structured, however, sometimes (long) sentence structures made it hard for me to follow. It would be nice, if the authors invest some time for rephrasing,giving the reader a more fluent reading experience.

The authors should use SI units in the preferred inverse notation throughout the manuscript (as stated in AMT manuscript guidelines), several times ppm-m is used (in the text and figure labels), which may be ppm m in SI units (?).

For better understanding, the authors should avoid synonyms (e.g. synthetic measure-ments vs. pseudo measurement vs. synthetic observation).

Generally, many of the figure legends, axis labels, and other labels might be to small for a good reproduction in the final publication.

Please, do not forget to add the necessary sections: "Data and code availability","Author contributions", "Conflict of interest".

I recommend publication in AMT, subject to some improvements.

We thank the reviewer for the constructive comments and appreciate the thoughtful review. Please find a point by point response below:

**Specific comments**

p. 3, ll. 6ff.:The inclusion of retrieval noise in the simulation of synthetic measure-ments is mentioned, but I have not found anything about this topic in Sec. 4.1 or Sec.5, which describe more details. Is additional noise used (and if, which) in preparing the synthetic measurements? Is this affecting the estimated errors (and how)? Maybe more details can be included in Section 4.2.

We performed experiments to add noise to the scenes using random Gaussian noise and found that the IME only changes by 7%, well within the 1-SD error from snapshot average. So we did not include the retrieval noise in our figures (but will consider this in the future when we will focus more on real data for which proper treatment of noise and plume segmentation will become increasingly important).

p. 3, l. 26: A 15 minute revisit time is mentioned, sub-figures of Fig. 1 show instrument overpasses in time intervals of 7 to 9 minutes (as also stated in the figure caption).
The revisit time is indeed 7-9 minutes apart. We have corrected the details in the figure caption to match with this.

p. 6: I recommend to restructure Section 4, as Section 4 itself is empty. It could rather be: "4 Large Eddy Simulation" and "5 Synthetic measurement", following Sections change accordingly. The section on synthetic measurement could be extended by some details on the additional noise added. Maybe the applied detection thresholds could also be included here, instead of in Section 2.
Based on the reviewer's suggestion, we adjusted the structure of the paper accordingly, and changed the number in the texts that mention to these sections.

p.6, l. 14: In this study latent and sensible surface heat fluxes are kept constant. Was the method tested with other settings (except of the controlled release exper-iment)? What would happen if these fluxes are varied? How would the plume be affected? Would this impact the method? If the method is still applicable with the derived correlations, would the error estimates change?Some of these questions are answered in "Discussion and conclusion" and answers may not fit in "LES setup". The authors should consider a reference to the discussion section and a strengthening of the corresponding paragraph there, or an additional section on limitations.

These values are based on typical field campaign data in the Four Corners area. To show how the result from our method applies to the field of different conditions, we have added additional LES runs with different combination of sensible and latent heat fluxes (SH and LH) in two more cases: SH = LH, LH > SH to compared with the typical condition we use in this paper (SH > LH)

[Figure]

We found that the relationship between observed IME, flux rate and wind speed under new conditions (orange and blue lines) lie within 1-SD error from our original condition (green line). This shows that the uncertainties associated with the change in these conditions will not significantly impacted our method and are captured well with the range of errors we have analysed.

We added a small section to show the relatively small impact in the discussion.

p.8, ll. 4ff.:If I understand it right, the instantaneous value for U-10 is written out from the LES simulation every minute. When the plume structure is – as stated in line 7 – influenced by the wind during this minute, the LES model (integration)time-step should be smaller. The model time-step should be somewhere around few seconds, taking into account the high resolution of 5 m. However, I found no value for the model time-step, maybe you could include it in the Section about the LESsetup. Maybe a rephrasing of the sentence in ll. 4f. could help to understand that instantaneous wind values are written out from the model simulation. Also, using the term "timestep" for one instance of the model output may be a bit misleading, as output does not coincide with every model time-step, maybe you could use "snapshot".

You are right, the model time-step is much shorter, namely one second. We added "The model computational time-step is one second." to the LES Set Up section. We paraphrased and adjusted the world "time-step" to "snapshots" accordingly.

p.9, ll. 17ff.:For the method to work, has the plume origin to be known for the angular mass binning? For LES simulation and field campaigns this should be no problem (when measuring unknown flux rates from a known point source). Also, for flux inversions of known sources from future high resolution satellite imagery, e.g, foremission monitoring, the method will be useful. In other situations, it might be more challenging. Limitations of the method should be addressed in the discussion section,or an additional section on limitations. (e.g., more than one point source or parts of a second plume included in a scene ...)

For large enough plumes, the plume origin can be easily found from the imagery itself. In fact, we will work on image classification schemes to automate this procedure in the future. However, in the current study we don't consider the plume origin to be a major factor. For HyTES data it might be as the origin cannot be readily inferred but with total column data in the short-wave infrared, the error in the origin is rather small.

p.10, ll. 16f.:The paragraph about error estimates could be strengthened. Please do not just give the value of$\chi 2$ without interpreting this result. Maybe additional description how the "average percentage error" is calculated, and how the error propagates through all steps of the method could be added.

We clarified that the average percentage error is the average of the percentage differences (in absolute terms) between the predicted value and the actual value for single point source predictions. It is approximately 30%. The reduced chi-2 value is 3.84 which suggests that the error variance may tend to be slightly underestimated for this individual point source prediction. However, when calculating the aggregate prediction, the predicted value is different to the actual value within only less than 10%. We added additional information on this section in the paper. We have added additional explanations for how the errors propagation is calculated from Figure 6, 10 and 11.

p.10, ll. 26ff.:Additional information about the controlled release experiment would be nice. Maybe, include a figure of the scene, or a figure of the angular distribution (comparable to Fig. 8)...

We have added the scene observed by AVIRIS-NG in the controlled release experiment and more information on date, time, and location to the scene.

p. 12, ll. 3ff.:This paragraph addresses the limitations arising from using constant sensible and latent surface heat fluxes. The authors should consider to present the limitations in a separate section. The last sentence of the paragraph should be moved closer to the discussion of the constant surface fluxes. All limitations should be named and discussed.

We added more analysis for the runs with two additional cases. One being the sensible heat flux (SH) and latent heat flux (LH) are equal to 220 W/m2. Another case is when SH is 200 and LH is 400 W/m2. This is two different cases in contrast to the typical condition of SH being higher than LH (400 W/m2 and 40 W/ms respectively in our method). Our result shows that the relationship between IME, flux rate and wind speed is not significantly impacted by the large change in heat fluxes, as described in the response to p.6, l. 14 above.

p. 20, Fig. 8:The figure contains two black lines, which are not explained, I as-sume they denote something like the main axis of the plume.If the plume is already rotated, why is the maximum of the normalized angular IME distribution not at 0?
The two black lines denote an angular bin of 0.5 degree that sweeps through the 2D plume to construct the angular distribution. It is just for illustrative purposes. We have added the meaning of the two black lines to the caption of Figure 8. The plume is rotated based on an axis that is defined by an angle that separates plumes into 50/50 ratio, which can be slightly different from the angle of maximum density.

**Technical corrections**

We have adjusted our text according to the suggested technical corrections.

[revised manuscript text omitted]

Formatted ... [161]
Formatted ... [162]
Formatted ... [163]
Formatted ... [164]
Formatted ... [165]
Formatted ... [166]
Formatted ... [167]
Formatted ... [168]
Formatted ... [169]
Formatted ... [170]
Formatted ... [171]
Formatted ... [172]
Formatted ... [173]
Formatted ... [174]
Formatted ... [175]
Formatted ... [176]
Formatted ... [177]
Formatted ... [178]
Formatted ... [179]
Formatted ... [180]

monotonically decreasing cone width with respect to wind speed. Our choice of parameterization in Fig. 10 is an exponential fit, which adequately captures the present relationship without overfitting. This result illustrates that the cone width is a metric that can differentiate wind speeds based on using only the spatial distribution of the plume. This finding, together with the variation of IME with flux rate (Fig. 6), can therefore provide flux inversion without the need for ground measurements. The next section describes steps for estimating the flux rates and its associated uncertainties.

**6.3 Flux Inversion and Error Analysis**

Based on the IME and plume morphology of any given scene, we can estimate the flux rate. First, according to Fig. 6, for a given value of the IME observed in the scene, we can find what are possible range of fluxes for each wind speed from the lower and upper estimate of 1 standard deviation. We can then parameterize this relationship between the flux rate and the wind speed for this particular value of the IME. An example for the case of the observed IME of 50 kg is demonstrated in Fig. 11. Secondly, based on the spatial distribution of the plume in the scene, we can follow the procedure to construct the angular mass distribution. Based on Fig. 10, using an angular width measured from the plume, we can predict the wind speed from the fitted curve. The associated uncertainties of the wind speed are approximated by the lower and upper estimate of 1 standard deviation. We assume that by projecting a value of plume width onto the corresponding range of wind speeds within 1 standard deviation range, we obtain uncertainties for predicted wind speed that approximately represent 1 standard deviation error for the wind speed distribution. The wind speed and its uncertainty can hence be translated into the estimate of the mean flux rate as well as the corresponding uncertainties from the relationship of the flux rate and wind speed, as in Fig. 11.

With this approach, we selected 90 random snapshots with random prescribed flux rates and wind speeds. We predict the flux rate from the IME and the spatial distribution of each of plume scene and compare it to its actual prescribed value, as shown in Fig. 12. The average of the percentage differences (in absolute terms) between the predicted value and the actual value for single point source predictions is approximately 30%. The $\chi^2$ value from the predictions in Fig. 12 is 3.84 suggesting that the error variance may tend to be slightly underestimated for an individual point source prediction.

Nevertheless, the results shown in Fig. 12 demonstrates that this method permits estimation of total emission flux rate. Most importantly, accounting for non-linearities and variable wind speed helps to avoid systematic biases. Thus, the method employed here can minimize systematic errors that could be induced by assumptions on wind speed. To verify this point, we performed an aggregation analysis by bootstrapping 30 plumes out of 500 plumes of various flux rates and wind speeds, with 3000 repetitions. The sample size of 30 is chosen arbitrarily but is large enough to represent a situation for the estimation of total fluxes from a region. The comparison between the predicted and the actual total flux aggregated over 30 plumes is shown in Fig. 13. The predictions lie close to the actual aggregated fluxes, as demonstrated by the concentration of points near the 1-to-1 line in Fig. 13, implying that there are no significant systematic biases in our method. The mean of absolute differences from all these aggregates is 5.1% with a standard deviation of 3.9%, while the average of all differences (negative and positive) results in 2.9% with the standard deviation of 5.9%.

**Formatted** ... [182]

**Formatted** ... [181]

**Formatted** ... [183]

**Formatted** ... [184]

**Formatted** ... [185]

**Formatted** ... [186]

**Formatted** ... [187]

**Formatted** ... [189]

**Formatted** ... [191]

**Formatted** ... [188]

**Formatted** ... [192]

**Formatted** ... [193]

**Formatted** ... [194]

**Formatted** ... [195]

**Formatted** ... [197]

**Formatted** ... [198]

**Formatted** ... [199]

**Formatted** ... [200]

**Formatted** ... [201]

**Formatted** ... [202]

**Formatted** ... [203]

**Formatted** ... [204]

**Formatted** ... [205]

**Formatted** ... [206]

**Formatted** ... [207]

**Formatted** ... [208]

**Formatted** ... [209]

**Formatted** ... [210]

**Formatted** ... [211]

[revised manuscript text omitted]

---

## Author Response (AR2)

Response to Minor Revision

Minor revisions:
Fig 1, caption: indicate the spatial measurements' resolution in m.
Fig 15, caption: 220 and 220 W m-2, not 2200 W m-2.

Responses:
Fig1, caption: I have indicated that the spatial measurement resolution for AVIRIS-NG is 2.8 m, and for HyTES is 2.3 m.
Fig15, caption: I have corrected the value of heat flux to be 220 W m-2.